# ACMP: Allen-Cahn Message Passing with Attractive and Repulsive Forces for Graph Neural Networks

**Yuelin Wang**[*]
Shanghai Jiao Tong University
`sjtu_wyl@sjtu.edu.cn`

**Kai Yi**[*]
University of New South Wales
`kai.yi@unsw.edu.au`

**Xinliang Liu**[*]
King Abdullah University of Science and Technology
`xinliang.liu@kaust.edu.sa`

**Yu Guang Wang**[†]
Shanghai Jiao Tong University
Shanghai Artificial Intelligence Laboratory
`yuguang.wang@sjtu.edu.cn`

**Shi Jin**[†]
Shanghai Jiao Tong University
Shanghai Artificial Intelligence Laboratory
`shijin-m@sjtu.edu.cn`

## Abstract

Neural message passing is a basic feature extraction unit for graph-structured data considering neighboring node features in network propagation from one layer to the next. We model such process by an interacting particle system with attractive and repulsive forces and the Allen-Cahn force arising in the modeling of phase transition. The dynamics of the system is a reaction-diffusion process which can separate particles without blowing up. This induces an Allen-Cahn message passing (ACMP) for graph neural networks where the numerical iteration for the particle system solution constitutes the message passing propagation. ACMP which has a simple implementation with a neural ODE solver can propel the network depth up to one hundred of layers with theoretically proven strictly positive lower bound of the Dirichlet energy. It thus provides a deep model of GNNs circumventing the common GNN problem of oversmoothing. GNNs with ACMP achieve state of the art performance for real-world node classification tasks on both homophilic and heterophilic datasets. Codes are available at `https://github.com/ykiiiiii/ACMP`.

## 1 Introduction

Graph neural networks (GNNs) have received a great attention in the past five years due to its powerful expressiveness for learning graph structured data, with broad applications from recommendation systems to drug and protein designs (Atz et al., 2021; Baek et al., 2021; Bronstein et al., 2021; 2017; Gainza et al., 2020; Wu et al., 2020). Neural message passing (Gilmer et al., 2017) serves as a fundamental feature extraction unit for graph-structured data that aggregates the features of neighbors in network propagation. We develop a GNN message passing, called the *Allen-Cahn message passing (ACMP)*, using interacting particle dynamics, where nodes are particles and edges representing the interactions of particles. The system is driven by both attractive and repulsive forces, plus the Allen-Cahn double-well potential from phase transition modeling. This model is motivated by the behavior of the particle system of collective behaviors common in nature and human society, for example, insects forming swarms to work; birds forming flocks to immigrate; humans forming parties

---

[*]equal contribution
[†]corresponding author

Figure 1: An illustration for one-step ACMP. Graph $\mathcal{G}_t$ with features $\mathbf{x}(t)$ in the purple and green blocks have different treatment of attraction or repulsion. The same color indicates similar node features. The node $\mathbf{x}(t)$ is updated by one step to $\mathbf{x}(t + \Delta t)$ via ODE solver. Nodes in the green block tend to attract each other and in the other block, nodes in different colors repel each other, and thus both colors are strengthened during propagation. It gives rise to forming bi-cluster flocking. The double-well potential turns features darker under gradient flow to circumvent blowup of energy.

to express public opinions. Various mathematical models have been proposed to model these behaviors (Albi et al., 2019; Motsch & Tadmor, 2011; Castellano et al., 2009; Proskurnikov & Tempo, 2017; Degond & Motsch, 2008). There are two major components in this model. First, while the attractive force forces all particles into one cluster, the repulsive forces allow particles to separate into two different clusters, which is essential to avoid oversmoothing. However, repulsive forces could make the Dirichlet energy diverge. We augment the model with the Allen-Cahn (Allen & Cahn, 1979) term (or Rayleigh friction (Rayleigh, 1894)), which is crucial in preventing the Dirichlet energy in the evolution from becoming unbounded, allowing us to prove mathematically that the lower bound of the Dirichlet energy is strictly bigger than zero, hence avoiding oversmoothing. Specifically, we will prove that under suitable conditions on the parameters, the dynamics of the ACMP particle system will time-asymptotically form $2^d$ different clusters and the Dirichlet energy has a strictly positive lower bound.

The structure of ACMP can handle two problems in GNNs: oversmoothing and heterophily. Oversmoothing (Nt & Maehara, 2019; Oono & Suzuki, 2019; Konstantin Rusch et al., 2022) means that all node features become undistinguishable, and equivalently, in the formulation of particle systems, features form only one consensus. Heterophily problems means GNNs perform worse in heterophilic graphs (Lim et al., 2021; Yan et al., 2021). It is due to the neighboring nodes of different classes are mistaken for the same class in GNNs like GCN and GAT. However, the presence of repulsion in ACMP makes particles separate into two different clusters, hence provides a simple and neat solution for prediction tasks on both two problems.

Overall, the benefit of the Allen-Cahn message passing with repulsion is manifold. 1) It circumvents oversmoothing issue, namely the Dirichlet energy is bounded from below. 2) The network is stable in the sense that features and Dirichlet energy are bounded from above. 3) Feature smoothness (energy decreasing) and the balance between nodes features and edge features can be adjusted by network parameters that control the attraction, repulsion and phase transition. The model can then reach an acceptable trade-off on self-features and neighbor effect, as shown in Figure 1. Our model can thus handle node classification tasks for both homophilic and heterophilic datasets by using only one-hop neighbour information. 4) The proposed model can be implemented by neural ODE solvers for the system with attractive and repulsive forces.

In theory, we prove that Dirichlet energy of GNNs with ACMP has a lower bound above zero (limiting oversmoothing), as well as an upper bound (circumventing blow-up) under specific conditions. This agrees with the experimental results (Section 6). We also prove that ACMP is a process for the features to generate clusters thanks to the double-well potential, which provides an interpretable theory for node classification.

## 2 BACKGROUND

**Message Passing in GNNs**  Graph neural networks are a kind of deep neural networks which take graph data as input. Neural Message Passing (MP) (Gilmer et al., 2017; Battaglia et al., 2018) is a most widely used propagator for node feature update in GNNs, which takes the following form: for

the undirected graph $\mathcal{G} = (\mathcal{V}, \mathcal{E})$ is with sets of nodes $\mathcal{V}$ and edges $\mathcal{E}$, with $\mathbf{x}_i^{(k-1)} \in \mathbb{R}^d$ denoting features of node $i$ in layer $(k-1)$ and $a_{j,i} \in \mathbb{R}^D$ edge features from node $j$ to node $i$,

$$\mathbf{x}_i^{(k)} = \gamma^{(k)} \left( \mathbf{x}_i^{(k-1)}, \square_{j \in \mathcal{N}_i} \, \phi^{(k)} \left( \mathbf{x}_i^{(k-1)}, \mathbf{x}_j^{(k-1)}, a_{j,i} \right) \right),$$

where $\square$ denotes a differentiable, (node) permutation invariant function, e.g., sum, mean or max, and $\gamma$ and $\phi$ denote differentiable functions such as MLPs (MultiLayer Perceptrons), and $\mathcal{N}_i$ is the set of one-hop neighbors of node $i$. The message passing updates the feature of each node by aggregating the self-feature with neighbors' features. Many GNN feature extraction modules such as GCN (Kipf & Welling, 2017), GAT (Veličković et al., 2018) and GIN (Xu et al., 2018) can be written as message passing. For example, the MP of GCNs reads, with learnable parameter matrix $\Theta$, $\mathbf{x}_i' = \Theta^\top \sum_{j \in \mathcal{N}_i \cup \{i\}} \frac{a_{j,i}}{\sqrt{\hat{d}_j \hat{d}_i}} \mathbf{x}_j$, where $\hat{d}_i = 1 + \sum_{j \in \mathcal{N}(i)} a_{j,i}$ and $\hat{D} = \text{diag}(\hat{d}_1, \ldots, \hat{d}_N)$ is the degree matrix for $A + I$. Graph attention network (GAT) uses attention coefficients $\alpha_{i,j}$ as similarity information between nodes in the MP update $\mathbf{x}_i' = \alpha_{i,i} \Theta \mathbf{x}_i + \sum_{j \in \mathcal{N}_i} \alpha_{i,j} \Theta \mathbf{x}_j$, with

$$\alpha_{i,j} = \frac{\exp \left( \text{LeakyReLU} \left( \mathbf{a}^\top [\Theta \mathbf{x}_i \, \| \, \Theta \mathbf{x}_j] \right) \right)}{\sum_{k \in \mathcal{N}_i \cup \{i\}} \exp \left( \text{LeakyReLU} \left( \mathbf{a}^\top [\Theta \mathbf{x}_i \, \| \, \Theta \mathbf{x}_k] \right) \right)}. \tag{1}$$

The MP framework was also developed as PDE solvers in Brandstetter et al. (2022b) by embedding differential equations as a parameter into message passing like Brandstetter et al. (2022a). This paper regards particle system evolution (ODE) as message passing propagation, and the appropriate design of the particle system offers desired properties for the resulting GNN.

**Graph neural diffusion**  Neural diffusion equations on graphs (GRAND) are proposed by Chamberlain et al. (2021), which provides a unified mathematical framework for some message passings:

$$\frac{\partial}{\partial t} \mathbf{x}(t) = \text{div}[\mathbf{G}(\mathbf{x}(t), t) \nabla \mathbf{x}(t)], \tag{2}$$

where $\mathbf{G} = \text{diag}(a(x_i(t), x_j(t), t))$ where $a$ is a function reflecting similarity between nodes $i$ and $j$, and $x_i$ is the scale-valued feature for node $i$, and $\mathbf{x} = \oplus x_i$.

## 3 MOTIVATIONS

### 3.1 ATTRACTIVE AND REPULSIVE FORCES

The equation (2) itself can be interpreted in a formulation different from diffusion. In this paper, we study the neural equations of *interacting particle system*, which has a similar structure to (2). We rewrite (2) into a component-wise version and obtain a particle system

$$\frac{\partial}{\partial t} x_i(t) = \sum_{j \in \mathcal{N}_i} a(x_i, x_j)(x_j - x_i). \tag{3}$$

In the formulation of particle systems, one can easily discover the evolution trend of the features. If $a(x_i, x_j) > 0$, the direction of $x_i$'s velocity is towards $x_j$, which means that $x_i$ is attracted by $x_j$. In the contrast, if $a(x_i, x_j) < 0$, $x_i$ has a trend to move away from $x_j$. Hence, $a(x_i, x_j)$ serves as the attractiveness or repulsiveness of the force between $x_i$ and $x_j$. In the diffusion model above, all $a(x_i, x_j)$'s are positive, therefore all the node features in one connected component attract each other. If the weight matrix $(a(x_i, x_j))_{N \times N}$ is right-stochastic, one can prove that the convex hull of the features will not dilate in time (see Motsch & Tadmor (2014); Chamberlain et al. (2021)). Such feature aggregation means that message propagates along the edges of the graph and some potential *consensus* forms in the process.

However, the message propagation does not limit to *consensus* (corresponding to diffusion). Information interaction can derive polarization of final judgement when *negative* message matters in some problems rather than *positive* message. For instance, in a node classification task on a bipartite, the neighbour message is negative since connected nodes belong to different classes. In the

formulation of particle systems, the mechanism of positive and negative messages can be modelled by adding bias $\beta_{i,j}$ into (3)

$$\frac{\partial}{\partial t}x_i(t) = \sum_{j \in \mathcal{N}_i}(a(x_i, x_j) - \beta_{i,j})(x_j - x_i). \tag{4}$$

The coefficient term $a(x_i, x_j) - \beta_{i,j}$ corresponds to the interactive force. By adjusting $\beta_{i,j}$, both in the system attractive and repulsive forces co-exist. If $a(x_i, x_j) - \beta_{i,j} > 0$, $x_i$ is attracted by $x_j$. While if $a(x_i, x_j) - \beta_{i,j} < 0$, $x_i$ is repelled by $x_j$. If the coefficient equates zero, there is no interaction between $x_i$ and $x_j$. Then, the dynamics is enabled to adapt both positive and negative message passing. In this way, the neural message passing can handle either homophilic or heterophilic datasets (see Section 6 for detailed discussion).

## 3.2 PSEUDO-GINZBURG-LANDAU ENERGY

However, adding the repulsion term may cause the particles being separated away infinitely, thus the Dirichlet energy becomes unbounded. To avoid this problem, we add a damping term $\delta x_i(1 - x_i^2)$, which we call an *Allen-Cahn term*. The coefficient $\alpha > 0$ is multiplied just for technical convenience.

$$\frac{\partial}{\partial t}x_i(t) = \alpha \sum_{j \in \mathcal{N}_i}(a(x_i, x_j) - \beta_{i,j})(x_j - x_i) + \delta x_i(1 - x_i^2). \tag{5}$$

**Gradient Flow** The variational principle governing many PDE models states that the equilibrium state is actually the minimizer of one specific energy. We first introduce the Dirichlet energy and show that (3) can be characterized by looking into the corresponding Euler-Lagrange equation of the Dirichlet energy. Let adjacent matrix $\mathbf{A}$ represent the undirected connectivity between nodes $x_i$ and $x_j$, with $a_{i,j} = 1$ for $(i,j) \in \mathcal{E}$ and $a_{i,j} = 0$ for $(i,j) \notin \mathcal{E}$. The *Dirichlet energy* $\mathbf{E}$ in terms of $\mathcal{G} = (\mathcal{V}, \mathcal{E})$ and node features $\mathbf{x} \in \mathbb{R}^{N \times d}$ takes the form

$$\mathbf{E}(\mathbf{x}) = \frac{1}{N}\sum_{i \in \mathcal{V}}\sum_{j \in \mathcal{N}_i}a_{i,j}\|\mathbf{x}_i - \mathbf{x}_j\|^2. \tag{6}$$

By calculus of variation, we can formulate the corresponding particle equation

$$\frac{\partial \mathbf{x}}{\partial t} = -\nabla_{\mathbf{x}}\mathbf{E}, \quad \frac{\partial x_i}{\partial t} = -\frac{\partial \mathbf{E}}{\partial x_i} = \frac{2}{N}\sum_{j \in \mathcal{N}_i}a_{i,j}(x_j - x_i). \tag{7}$$

On the RHS of (7), the summation takes over the one-hop neighbors $\mathcal{N}_i$ of node $i$, which aggregates the impact from the neighboring nodes. Equation (7) is (5) when one takes adjacent matrix $\mathbf{A}$ as the weight matrix $(a(x_i, x_j))_{N \times N}$.

**Particle equation with the double-well potential** To avoid blowing-up of the solution, one can design an external potential to control the solutions so they are bounded. Here, we define the *pseudo-Ginzburg-Landau energy* on graph $\mathcal{G}$ denoted by $\Phi : L^2(\mathcal{V}) \to \mathbb{R}$, as a combination of the *interacting energy* and *double-well potential* $W : \mathbb{R} \to \mathbb{R}_+$, with

$$W(x) = (\delta/4)(1 - \|x\|^2)^2, \qquad \Phi(\mathbf{x}) = \frac{1}{2}\alpha \sum_{i \in \mathcal{V}}\sum_{j \in \mathcal{N}_i}(a_{i,j} - \beta_{i,j})\|x_i - x_j\|^2 + \sum_{i \in \mathcal{V}}W(x_i), \tag{8}$$

where parameters $\alpha, \delta > 0$ are used to balance the two types of energy. From now on, we denote $a(x_i, x_j)$ by $a_{i,j}$ for simplicity. The pseudo-Ginzburg-Landau energy is not a true energy because the matrix $(a_{i,j} - \beta_{i,j})_{N \times N}$ can be non-positive definite. If $\beta_{i,j}$'s all equate zero, it then becomes the Ginzburg-Landau energy defined in Bertozzi & Flenner (2012); Luo & Bertozzi (2017). Using this combined energy, we can obtain the Allen-Cahn equation with repulsion on graph as $\frac{\partial \mathbf{x}}{\partial t} = -\nabla_{\mathbf{x}}\Phi$, which is equivalent to (5).

## 4 ALLEN-CAHN MESSAGE PASSING

We propose the *Allen-Cahn Message Passing* (ACMP) neural network based on equation (5), where the message is updated by the evolution of the equation via a neural ODE solver. To our best knowledge, this is the *first time* to introduce a type of message passing to amplify the difference between connected nodes by repulsive force.

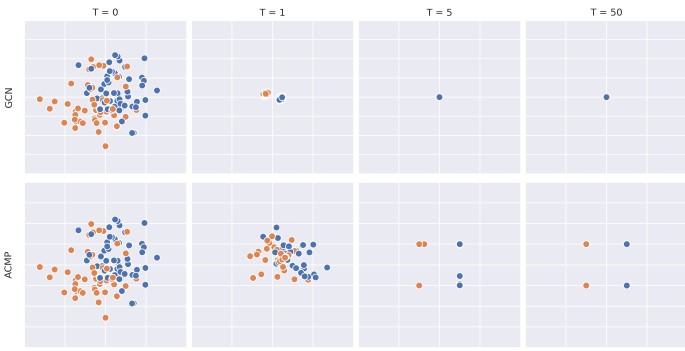

Figure 2: We compare the evolution of node features in GCN and ACMP. The initial position is represented by the 2-dimensional position of the nodes, which is shown in the first column. The GCN aggregates all node features by taking the weighted average of its neighbors' features. With the propagated steps increasing, all the nodes' features shrink to a point, which gives rise to over-smoothing. When it comes to ACMP, nodes' features are grouped by four attractors, which helps to circumvent oversmoothing. More details can be seen in Section 6.

**Network Architecture** Suppose $d$-dimensional node-wise features represented by a matrix $\mathbf{x}^{\text{in}}$ where row $i$ represents feature of node $i$. Our scheme first embeds the node feature $\mathbf{x}(0) = \text{MLP}(\mathbf{x}^{\text{in}})$ by a simple multi-layer perceptron (MLP), which is treated as an input for ACMP propagation $\mathcal{A} : \mathbb{R}^d \to \mathbb{R}^d$, by $\mathbf{x}(0) \mapsto \mathbf{x}(T)$, where $\mathbf{x}(T) = \mathbf{x}(0) + \int_0^T \frac{\partial \mathbf{x}(t)}{\partial t} dt$, $\quad \mathbf{x}(0) = \text{MLP}\left(\mathbf{x}^{\text{in}}\right)$, where $\frac{\partial \mathbf{X}(t)}{\partial t}$ is estimated by ACMP defined on $\mathcal{G}$ based on (5). The node features $\mathbf{x}(T)$ at the ending time are fed into an MLP based classifier. Then, we define the Allen-Cahn message passing by

$$\frac{\partial}{\partial t}\mathbf{x}_i(t) = \boldsymbol{\alpha} \odot \sum_{j \in \mathcal{N}_i} (a(\mathbf{x}_i(t), \mathbf{x}_j(t)) - \beta)(\mathbf{x}_j(t) - \mathbf{x}_i(t)) + \boldsymbol{\delta} \odot \mathbf{x}_i(t) \odot (1 - \mathbf{x}_i(t) \odot \mathbf{x}_i(t)). \quad (9)$$

Here $\boldsymbol{\alpha}, \boldsymbol{\delta} \in \mathbb{R}^d$ are learnable vectors of the same length as the node feature $\mathbf{x}_i$. While we can use a more general case when each edge $(i, j)$ uses different trainable $\beta_{i,j}$, we have simplied to single hyper-parameter $\beta \in \mathbb{R}^+ \cup \{0\}$, which makes the network and optimization easier. The $\beta$ in our model is a crucial parameter, which can be adjusted such that the attractive and repulsive forces both present to enrich the message passing effect. If one chooses $\boldsymbol{\delta} = 0$, $\beta = 0$, our model is reduced to the graph neural diffusion network (GRAND) in Chamberlain et al. (2021). In experiments, we would make significant use of nontrivial $\boldsymbol{\delta}$ and $\beta$.

The operations of all terms are channel-wise, involving $d$ channels, except $a(\mathbf{x}_i(t), \mathbf{x}_j(t))$, and $\odot$ represents channel-wise multiplication for $d$ feature channels. Figure 1 illustrates the one-step ACMP mechanism (9): Nodes with close colors attracts each other otherwise repel. Nodes in the same block tend to attract each other and both colors are strengthened during message passing propagation. The double-well potential prevents the features and Dirichlet energy from blowup. In this process, node feature $\mathbf{x}(t)$ is updated to $\mathbf{x}(t + \Delta t)$ for a time increment $\Delta t$. Ultimately, a bi-cluster flock is formed for node classification.

In the propagation of ACMP in (9), we need to specify how the neighbors are interacted, that is how the $a(\mathbf{x}_i(t), \mathbf{x}_j(t))$ is evolved with time. There are many kinds of methods to update the edge weights. The two typical types are GCN (Kipf & Welling, 2017) and GAT (Veličković et al., 2018).

**ACMP-GCN**: this model uses deterministic $a(\mathbf{x}_i(t), \mathbf{x}_j(t))$, which is given by the adacency matrix $A = (a_{i,j})$ of the original input graph $\mathcal{G}$ and does not change with time. That is, the coefficients in GCNs $a_{i,j}^{\text{GCN}} := a_{i,j} / \sqrt{\hat{d}_i \hat{d}_j}$. The message passing of (9) is reduced to

$$\frac{\partial}{\partial t}\mathbf{x}_i(t) = \boldsymbol{\alpha} \odot \sum_{j \in \mathcal{N}_i} (a_{i,j}^{\text{GCN}} - \beta)(\mathbf{x}_j(t) - \mathbf{x}_i(t)) + \boldsymbol{\delta} \odot \mathbf{x}_i(t) \odot (1 - \mathbf{x}_i(t) \odot \mathbf{x}_i(t)). \quad (10)$$

**ACMP-GAT**: we can replace $a_{i,j}^{\mathrm{GCN}}$ in (10) by the attention coefficients (1) of GAT, which with extra trainable parameters measures the similarity between two nodes by taking account of both node and structure features. The system then drives edges to update in each iteration of message passing.

**Neural ODE Solver**   Our method uses an ODE solver to numerically solving the equation ((9) and (10)). To obtain the node features $\mathbf{x}(T)$, we need a stable numerical integrator for solving the ODE efficiently and backpropagation of gradients. Since our model is stable in terms of evolution time, most explicit and implicit numerical methods such as explicit Euler, Runge-Kutta 4th-order, midpoint, Dormand-Prince5 (Chen et al., 2018; Lu et al., 2018; Norcliffe et al., 2020; Chamberlain et al., 2021) work well as long as the step size $\tau$ is small enough. In experiments, we implement ACMP using Dormand-Prince5 method which provides a fast and stable numerical solver. The *network depth* of ACMP-GNN is equal to the numerical iteration number $n_t$ set in the solver.

**Computational Complexity**   The computational complexity of the ACMP is $\mathcal{O}(NEdn_t)$, where $n_t$, $N$, $E$ and $d$ are number of time steps in time interval $[0, T]$, number of nodes, number of edges and number of feature dimension, respectively. Since our model only considers nearest (one-hop) neighbors, $E$ is significantly smaller than that of graph rewiring (Gasteiger et al., 2019; Alon & Yahav, 2021) and multi-hop (Zhu et al., 2020) methods.

**Channel Mixer**   Channel mixing can be spontaneously introduced from the perspective of diffusion coefficients though our model is previously written in the channel-wise form. Whether channel mixing happens depends on the specific GNN driver we choose for ACMP. When the coefficients $\mathbf{a}(\mathbf{x}_i(t), \mathbf{x}_j(t))$ in (9) that do not update with time are a scalar or vector, like in ACMP-GCN, the operations of the message passing propagator are channel-wise and channel mixing is not incorporated. On the other hand, the ACMP-GAT with graph attention driver incorporates a learnable channel mixing when the coefficients are tensors. The channel mixer can be introduced by generalizing the Dirichlet energy to high dimension, for example, $\mathbf{E}(\mathbf{x}) := \frac{1}{N} \sum_{i \in \mathcal{V}} \sum_{j \in \mathcal{N}_i} (\mathbf{x}_i - \mathbf{x}_j)^T \mathbf{a}_{i,j} (\mathbf{x}_i - \mathbf{x}_j)$,

when $\mathbf{a}_{i,j} \in \mathbb{R}^{d \times d}$ are connectivity tensors.

## 5   DIRICHLET ENERGY

The dynamics (5) can circumvent the oversmoothing issue of GNNs (Nt & Maehara, 2019; Oono & Suzuki, 2019; Konstantin Rusch et al., 2022). Oversmoothing phenomenon means that all node features converge to the same constant – consensus forms – as the network deepens, and equivalently, the Dirichlet energy will decay to zero exponentially. This idea was first introduced in Cai & Wang (2020). Konstantin Rusch et al. (2022) gives an explicit form for oversmoothing.

In our model, as we will show below, the node features in each channel tend to evolve into two clusters departing from each other under certain conditions. This implies a strictly positive lower bound of the Dirichlet energy. In addition, the system will not blow up thanks to the Allen-Cahn term. We put all the proofs and some related supplementary results in the appendix.

**Proposition 1.** *If $\delta > 0$, the node features $\mathbf{x}_i$ in (5) is bounded in terms of $\|\cdot\|$ and energy for all $t > 0$, i.e., $\mathbf{E}(\mathbf{x}(t)) \leq C$, and $\|\mathbf{x}\| \leq C$, where the constant $C$ only depends on $N$ and $\lambda_{\max}$.*

In the following propositions, we imitate the emergent behavior analysis in Fang et al. (2019) (see Appendix for details). For a graph $\mathcal{G}$ with $N$ nodes, its vertices are said to form *bi-cluster flocking* if there exist two disjoint sets of vertex subsets $\{\mathbf{x}_i^{(1)}\}_{i=1}^{N_1}$ and $\{\mathbf{x}_i^{(2)}\}_{j=1}^{N_2}$ satisfying

$$
\begin{aligned}
&(i)\ \sup_{0 \leq t < \infty} \max_{1 \leq i,j \in N_1} |x_i^{(1)}(t) - x_j^{(1)}(t)| < \infty, \quad \sup_{0 \leq t < \infty} \max_{1 \leq i,j \in N_2} |x_i^{(2)}(t) - x_j^{(2)}(t)| < \infty; \\
&(ii)\ \exists\, C', T^{**} > 0 \text{ such that } \min_{1 \leq i \in N_1, 1 \leq j \in N_2} \{|x_i^{(1)}(t) - x_j^{(2)}(t)|\} \geq C', \quad \forall t > T^{**},
\end{aligned}
\tag{11}
$$

where $x_i^{(1)}, x_i^{(2)}$ denote any component of $\mathbf{x}_i^{(1)}, \mathbf{x}_i^{(2)}$.

We now show the long-time behaviour of model (5) following the analysis of Fang et al. (2019) for strength coupling $(\alpha, \delta)$ that satisfies the following condition: there exists $\{\beta_{i,j}\}$ such that $\mathcal{I} :=$

$\{1, \ldots, N\}$ can be divided into two disjoint groups $\mathcal{I}_1, \mathcal{I}_2$ with $N_1$ and $N_2$ particles respectively:

$$
\begin{aligned}
0 < S \leq \overline{a}_{i,j} \quad \text{with } \overline{a}_{i,j} := a_{i,j} - \beta_{i,j} && \text{for } i, j \in \mathcal{I}_1, \\
0 < S \leq \overline{a}_{i,j} \quad \text{with } \overline{a}_{i,j} := a_{i,j} - \beta_{i,j} && \text{for } i, j \in \mathcal{I}_2, \\
0 \leq \overline{a}_{i,j} \leq D \quad \text{with } \overline{a}_{i,j} := -(a_{i,j} - \beta_{i,j}) && \text{otherwise,}
\end{aligned}
\tag{12}
$$

where $S, D$ are independent of time $t$. The $S$ and $D$ in (12) are the repulsive and attractive forces. We prove that if the repulsive force between the particles is stronger than the attractive force, that is, $S > D$, the system is guaranteed to have bi-cluster flocking, as shown in Proposition 2 below. For time $t \geq 0$, suppose $\mathbf{x}_c^{(1)}(t)$ and $\mathbf{x}_c^{(2)}(t)$ are the *feature centers* of the two groups of the particles $\{\mathbf{x}_i^{(1)}(t)\}_{i=1}^{N_1}$ and $\{\mathbf{x}_j^{(2)}(t)\}_{j=1}^{N_2}$ which are partitioned as above from the whole vertex set $\mathcal{V}$, given by

$$
\mathbf{x}_c^{(1)}(t) := \frac{1}{N_1} \sum_{i=1}^{N_1} \mathbf{x}_i^{(1)}(t), \quad \mathbf{x}_c^{(2)}(t) := \frac{1}{N_2} \sum_{i=1}^{N_2} \mathbf{x}_i^{(2)}(t).
$$

Suppose $\mathbf{x}_c^{(s)}(t)$ has the $d$-dimensional feature, and let $\mathbf{x}_{c,k}^{(s)}(t)$, $k = 1, \ldots, d$, be the $k^{th}$ (dimension) component of the feature $\mathbf{x}_c^{(s)}(t)$, $s = 1, 2$.

**Proposition 2.** *The system (5) has a bi-cluster flocking if for each $k = 1, \ldots, d$, the initial $|\mathbf{x}_{c,k}^{(1)}(0) - \mathbf{x}_{c,k}^{(2)}(0)| \gg 1$, and if there exists a positive constant $\eta$ such that*

$$
\alpha(S - D) \min\{N_1, N_2\} \geq \delta + \eta,
\tag{13}
$$

*where the $\delta$ is the weight factor for the double-well potential in the equation (5).*

**Proposition 3.** *For system (5) with bi-cluster flocking, there exists a constant $C > 0$ and some time $T^*$ such that $\forall t \geq T^*$,*

$$
|\mathbf{x}_i^{(1)}(t) - \mathbf{x}_j^{(2)}(t)| \geq C > 0, \quad \forall i, j.
$$

*Thus, if the non-zero $a_{i,j}$ are all positive, the Dirichlet energy for ACMP is lower bounded by a positive constant.*

## 6 EXPERIMENTS

**Dirichlet Energy** We first illustrate the evolution of the Dirichlet energy of ACMP by an undirected synthetic random graph. The synthetic graph has 100 nodes with two classes and 2D feature which is sampled from the normal distribution with the same standard deviation $\sigma = 2$ and two means $\mu_1 = -0.5$, $\mu_2 = 0.5$. The nodes are connected randomly with probability $p = 0.9$ if they are in the same class, otherwise nodes in different classes are connected with probability $p = 0.1$. We compare the performance of GNN models with four message passing propagators: GCNs (Kipf & Welling, 2017), GAT (Veličković et al., 2018), GRAND (Chamberlain et al., 2021) and ACMP-GCN. In Figure 2, we visualize how the node features evolve from their initial state to their final steady state when 50 layers of GNN are applied. Additionally, in Figure 3, we show the Dirichlet energy of each layer's output in logarithm scales. Traditional GNNs such as GCNs and GAT suffer oversmoothing as the Dirichlet energy exponentially decays to zero in the first ten layers. GRAND relieves this problem by multiplying a small constant which can delay all nodes' features to collapse to the same value. For ACMP, the energy stabilizes at the level that relies upon the roots of the double-well potential in (9) after slightly decaying in the first two layers.

**Node Classification** We compare the performance of ACMP with several popular GNN model architectures on various node classification benchmarks, containing both homophilic and heterophilic datasets. Graph data is considered as *homophilic* (Pei et al., 2020) if similar nodes in the graph tend to connect together. Conversely, the graph data is said *heterophilic* if it has a small homophily level, when most neighbors do not have the same label with source nodes. We aim to demonstrate that ACMP is a flexible GNN model which can learn well both kinds of datasets by balancing the attractive and repulsive forces. The GCN for examples cannot perform well for heterophilic dataset as its message passing aggregates only the neighbor (1-hop) nodes. The neural ODE is solved by `Torchdiffeq` package with Dormand–Prince adaptive step size scheme. Only few hyperparameters are needed to be tuned in our model. For all the experiments, we fine tune the learning rate, weight decay, dropout, hidden dimensional of the $\beta$ which controls the repulsive force between nodes. We outline the details of hyperparameter search space in Appendix D.

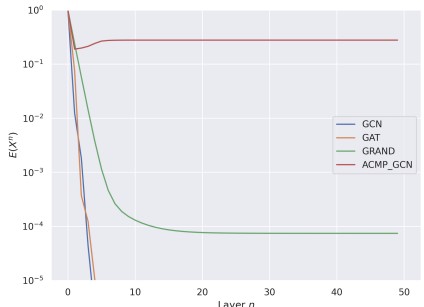
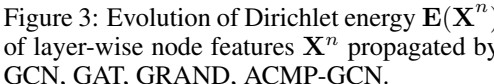
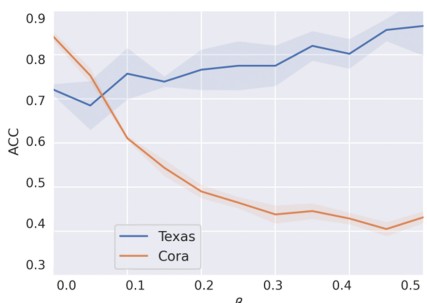

Figure 3: Evolution of Dirichlet energy $\mathbf{E}(\mathbf{X}^n)$ of layer-wise node features $\mathbf{X}^n$ propagated by GCN, GAT, GRAND, ACMP-GCN.

Figure 4: Significance plot for $\beta$ in terms of test accuracy on Cora (orange) and Texas (blue) with 10 fixed random splits.

**Homophilic datasets**  Our results are presented for the most widely used citation networks: Cora (McCallum et al., 2000), Citeseer (Sen et al., 2008) and Pubmed (Namata et al., 2012). Moreover, we evaluate our model on the Amazon co-purchasing graphs Computer and Photo (Namata et al., 2012), and CoauthorCS (Shchur et al., 2018). We compare our model with traditional GNN models: Graph Convolutional Network (GCN) (Kipf & Welling, 2017), Graph Attention Network (GAT) (Veličković et al., 2018), Mixture Model Networks (Monti et al., 2017) and GraphSage (Hamilton et al., 2017). We also compare our results with recent ODE-based GNNs, Continuous Graph Neural Networks (CGNN) (Xhonneux et al., 2020), Graph Neural Ordinary Differential Equations (GDE) (Poli et al., 2020) and Graph Neural Diffusion (GRAND) (Chamberlain et al., 2021). To address the limitations of this evaluation methodology proposed by Shchur et al. (2018), we report results for all datasets using 100 random splits with 10 random initialization's, and show the results in Table 1.

Table 1: Test accuracy and std for 10 initialization and 100 random train-val-test splits on six node classification benchmarks. **Red** (First), **blue** (Second), and **violet** (Third) are the best three methods.

| Random Split
Homophily level | **Cora**
**0.83** | **CiteSeer**
**0.71** | **PubMed**
**0.79** | **Coauthor CS**
**0.80** | **Computer**
**0.77** | **Photo**
**0.83** |
|---|---|---|---|---|---|---|
| **GCN** (Kipf & Welling, 2017) | $81.5 \pm 1.3$ | $71.9 \pm 1.9$ | $77.8 \pm 2.9$ | $91.1 \pm 0.5$ | $82.6 \pm 2.4$ | $91.2 \pm 1.2$ |
| **GAT** (Veličković et al., 2018) | $81.8 \pm 1.3$ | $71.4 \pm 1.9$ | $78.7 \pm 2.3$ | $90.5 \pm 0.6$ | $78.0$ | $85.7$ |
| **GAT-ppr** (Veličković et al., 2018) | $81.6 \pm 0.3$ | $68.5 \pm 0.2$ | $76.7 \pm 0.3$ | $91.3 \pm 0.1$ | $85.4 \pm 0.1$ | $90.9 \pm 0.3$ |
| **MoNet** (Monti et al., 2017) | $81.3 \pm 1.3$ | $71.2 \pm 2.0$ | $78.6 \pm 2.3$ | $90.8 \pm 0.6$ | $83.5 \pm 2.2$ | $91.2 \pm 2.3$ |
| **GraphSage-mean** (Hamilton et al., 2017) | $79.2 \pm 7.7$ | $71.6 \pm 2.0$ | $77.4 \pm 2.2$ | $91.3 \pm 2.8$ | $82.4 \pm 1.8$ | $91.4 \pm 1.3$ |
| **GraphSage-max** (Hamilton et al., 2017) | $76.6 \pm 1.9$ | $67.5 \pm 2.3$ | $76.1 \pm 2.3$ | $85.0 \pm 1.1$ | N/A | $90.4 \pm 1.3$ |
| **CGNN** (Xhonneux et al., 2020) | $81.4 \pm 1.6$ | $66.9 \pm 1.8$ | $66.6 \pm 4.4$ | $92.3 \pm 0.2$ | $80.29 \pm 2.0$ | $91.39 \pm 1.5$ |
| **GDE** (Poli et al., 2020) | $78.7 \pm 2.2$ | $71.8 \pm 1.1$ | $73.9 \pm 3.7$ | $91.6 \pm 0.1$ | $81.9 \pm 0.6$ | $92.4 \pm 2.0$ |
| **GRAND-l** (Chamberlain et al., 2021) | $83.6 \pm 1.0$ | $73.4 \pm 0.5$ | $78.8 \pm 1.7$ | $92.9 \pm 0.4$ | $83.7 \pm 1.2$ | $92.3 \pm 0.9$ |
| **ACMP-GCN** (ours) | $84.9 \pm 0.6$ | $75.0 \pm 1.0$ | $78.9 \pm 1.0$ | $93.0 \pm 0.5$ | $83.5 \pm 1.4$ | $91.8 \pm 1.1$ |
| **ACMP-GAT** (ours) | $82.3 \pm 0.5$ | $75.5 \pm 1.0$ | $79.4 \pm 0.4$ | $91.8 \pm 0.1$ | $84.4 \pm 1.6$ | $91.1 \pm 0.7$ |

**Heterophilic datasets**  We evaluate ACMP-GCN on the heterophilic graphs; Cornell, Texas and Wisconsin from the WebKB dataset[1]. In this case, the assumption of common neighbors does not hold. The poor performance of GCN and GAT models shown in Table 2 indicates that many GNN models struggle in this setting. Introducing repulsion can improve the performance of GNNs on heteroplilic datasets significantly. ACMP-GCN scores 30% higher than the original GCN for the Texas dataset which has the smallest homophily level among the datasets in the table.

**Attractive and Repulsive interpretation**  As shown in Table 2 and Table 1, ACMP-GCN and ACMP-GAT achieve better performance than GCN and GAT on both homophilic and heterophilic datasets. The majority of $a_{i,j} - \beta$ in the homophilic are positive, which means most nodes are attracted to each other. Conversely, most $a_{i,j} - \beta$ for the heterophilic are negative, which means that most nodes are repelled by their neighbors. Several GNNs exploiting multi-hop information can

---

[1]http://www.cs.cmu.edu/afs/cs.cmu.edu/project/theo-11/www/wwkb/

Table 2: Node classification results on heterophilic datasets. We use the 10 fixed splits for training, validation and test from Pei et al. (2020) and show the mean and std of test accuracy. **Red** (First), **blue** (Second), and **violet** (Third) are the best three methods.

| Homophily level | Texas 0.11 | Wisconsin 0.21 | Cornell 0.30 |
|---|---|---|---|
| **GPRGNN** (Chien et al., 2021) | $78.4 \pm 4.4$ | $82.9 \pm 4.2$ | $80.3 \pm 8.1$ |
| **H2GCN** (Zhu et al., 2020) | $84.9 \pm 7.2$ | $87.7 \pm 5.0$ | $82.7 \pm 5.3$ |
| **GCNII** (Chen et al., 2020) | $77.6 \pm 3.8$ | $80.4 \pm 3.4$ | $77.9 \pm 3.8$ |
| **Geom-GCN** (Pei et al., 2020) | $66.8 \pm 2.7$ | $64.5 \pm 3.7$ | $60.5 \pm 3.7$ |
| **PairNorm** (Zhao & Akoglu, 2020) | $60.3 \pm 4.3$ | $48.4 \pm 6.1$ | $58.9 \pm 3.2$ |
| **GraphSAGE** (Hamilton et al., 2017) | $82.4 \pm 6.1$ | $81.2 \pm 5.6$ | $76.0 \pm 5.0$ |
| **MLP** | $80.8 \pm 4.8$ | $85.3 \pm 3.3$ | $81.9 \pm 6.4$ |
| **GAT** (Veličković et al., 2018) | $52.2 \pm 6.6$ | $49.4 \pm 4.1$ | $61.9 \pm 5.1$ |
| **GCN** (Kipf & Welling, 2017) | $55.1 \pm 5.2$ | $51.8 \pm 3.1$ | $60.5 \pm 5.3$ |
| **GraphCON** (Konstantin Rusch et al., 2022) | $85.4 \pm 4.2$ | $87.8 \pm 3.3$ | $84.3 \pm 4.8$ |
| **ACMP-GCN** (ours) | $86.2 \pm 3.0$ | $86.1 \pm 4.0$ | $85.4 \pm 7.0$ |

achieve high performance in node classification (Zhu et al., 2020; Luan et al., 2021). However, high-order neighbor information will make the adjacency matrix dense and therefore can not be extended to large graphs, due to heavier computational cost. In our model, we take only one-hop information into account and add repulsive force ($\beta \geq 0$) to message passing, which has achieved the same or higher level of accuracy as multi-hop models in heterophilic datasets.

**Performance of ACMP to $\beta$** Hyperparameter $\beta$ is a signal of the repulsive force, meaning that when $a_{ij} - \beta$ is negative, the two nodes repel one another. To illustrate $\beta$'s impact, we use GCN as a diffusion term as $a_{ij}$ do not change during the ODE process and all the changes are related to $\beta$. As shown by Figure 4, ACMP performs best in Cora (orange curve) when all nodes are attracted to one another i.e., all $a_{ij} - \beta$ is positive. As the beta increases, the performance of the model degrades. In contrast, for the Texas dataset, when all force is attractive, ACMP achieves only 70% accuracy (blue curve). As $\beta$ increases, most $a_{ij} - \beta$ is negative, and the model's performance gets better.

## 7 RELATED WORK

**Neural differential equations** The topic of neural ODEs becomes an emerging field since E (2017) and Chen et al. (2018), with many follow-up works in the GNN field (Avelar et al., 2019; Poli et al., 2020; Sanchez-Gonzalez et al., 2019). GRAND (Chamberlain et al., 2021) propagated GNNs by the graph diffusion equation and Wu et al. (2023) developed an energy-constrained diffusion transformer. GraphCON (Konstantin Rusch et al., 2022) employed a second-order system to conquer oversmoothing of deep graph neural networks. By exploiting the fixed point of the dynamical system, Gallicchio & Micheli (2020) proposed FDGNN as an approach to graph classification.

**Allen-Cahn based variational graph models** In Bertozzi & Flenner (2012); Luo & Bertozzi (2017); Merkurjev et al. (2013) and references therein, authors extended Allen-Cahn related potential to graphical framework and developed a class of variational algorithms to solve the clustering, semisupervised learning and graph cutting problems. The new ingredient of graph neural network which enables us to combine learnable attraction and repulsion separates our method from the classical variational graph models.

## 8 CONCLUSION

We develop a new message passing method with simple implementation. The method is based on the Allen-Cahn particle system with repulsive force. The proposed ACMP inherits the characteristic dynamics of the particle system and thus shows adaption for node classification tasks with high homophily difficulty. Also, it propels networks to dozens of layers without getting oversmoothing. A strictly positive lower bound of the Dirichlet energy is shown by theoretical and experimental results which guarantees non-oversmoothing of ACMP. Experiments show excellent performance of the model for various real datasets.

## ACKNOWLEDGEMENTS

This work was supported by the Shanghai Municipal Science and Technology Major Project, and Science and Technology Commission of Shanghai Municipality grant No. 20JC1414100, (2021SHZDZX0102), and Shanghai Artificial Intelligence Laboratory (P22KN00524). S. Jin was partially supported by the NSFC grant No. 12031013.

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

## A    FLOCKING AND CONSENSUS

The microscopic (agent-based particle systems) modeling of flocking and consensus has been extensively studied. Motsch & Tadmor (2014) reviews a general class of models for self-organized dynamics and shows the relationship between heterophily and conseus. Castellano et al. (2009) presents a series of social and dynamics under the formulation of statistical physics. The flocking problem is to some degree similar to the general consensus problem (Olfati-Saber et al., 2007) which studies the emergent behaviours for multi-agent systems. The Cucker-Smale (in short C-S) model (Cucker & Smale, 2007) is a famous model in this field considering a second-order system adopting to classical dynamics. Ha & Tadmor (2008) and Ha et al. (2010) discuss asymptotic flocking for the C-S model with the Rayleigh friction. Fang et al. (2019) furthermore studies frameworks leading to bi-cluster flocking for the C-S model with the Raleigh friction and attractive-repulsive coupling.

## B    MODEL VARIANTS

**More clusters**   We can simply replace the double well potential $W$ by a multi-well potential to generate more equilibria. We provide two alternatives here. One can use a higher-order polynomial to construct additional wells. In general, a $(2k + 1)^{th}$ order polynomial can produce $k + 1$ stable equilibria in a proper form, which gives rise to more stable clusters. One can also use $\sin((\frac{3}{2} + l)\pi x + \frac{\pi}{2})$, $l = 0, \cdots, k$, defined on the interval $[-1, 1]$ as the multi-well potential, which has $l + 2$ stable equilibria.

**Stronger trapping force**   As the consensus state (i.e., $x_i = x_j$ for all $i, j$) might not be a global equilibrium of (10), particles could escape from one well of the potential of $W$ to another well. We can circumvent this instability by enhancing the attraction of the wells, which can be achieved by reducing the diffusion power around wells:

$$\frac{\partial}{\partial t}\mathbf{x}_i(t) = \boldsymbol{\alpha} \odot \sum_{j \in \mathcal{N}_i} (a^{\text{GNN}}(\mathbf{x}_i(t), \mathbf{x}_j(t)) - \beta)(\mathbf{x}_j(t) - \mathbf{x}_i(t))\left(1 - \mathbf{x}_i(t)^{\odot 2}\right)^{\odot 2} + \boldsymbol{\delta} \odot \mathbf{x}_i(t) \odot \left(1 - \mathbf{x}_i(t)^{\odot 2}\right).$$
(14)

where 'GNN' in $a^{\text{GNN}}$ can be GCN or attn, and $z^{\odot 2}$ is $z \odot z$. With this modification in (14), in any channel $k$, if any particle $\mathbf{x}_i^{(k)}$ gets caught in one potential well, then it is not likely to escape:

**Proposition 4.** *For (14), there exists a proper $\delta' > 0$ such that $\mathbf{x}_i^{(k)} \in [-1, -1 + \delta') \cup (1 - \delta', 1]$, then particle $\mathbf{x}_i^{(k)}$ cannot transition into another well.*

**Proof.** For the $\beta = 0$ case, assume $x_i = -1 + \epsilon$ for $\epsilon \leq \delta' < 1$ at a certain time $t_0$, that is, $x_i \in [-1, -1 + \delta')$. We want to show $\frac{dx_i}{dt}\big|_{t=t_0} < 0$, which means

$$\alpha \sum_{j \in \mathcal{N}_i} a_{i,j}(x_j - x_i)(1 - x_i^2)^2 < -\delta x_i(1 - x_i^2).$$

By $\sum_{j \in \mathcal{N}_i} a_{i,j} = 1$ from (26), the above inequality is equivalent to

$$\sum_{j \in \mathcal{N}_i} a_{i,j}x_j < \frac{\delta}{\alpha}\frac{1 - \epsilon}{2 - \epsilon}\frac{1}{\epsilon} + \epsilon - 1 \leq \frac{\delta}{2\alpha}\frac{1}{\epsilon} + \epsilon - 1 \leq \frac{\delta}{2\alpha\epsilon}.$$
(15)

Since $\{x_j\}_{j=1}^N$ are bounded (See Proposition 1.), (15) is satisfied for a sufficiently small $\delta'$. The other case $x_i = 1 - \epsilon$ can be similarly proved.

For the $\beta \neq 0$ case, we also assume $x_i = -1 + \epsilon$ for $\epsilon \leq \delta' < 1$ at a certain time $t_0$. Similarly with (15), we have

$$\sum_{j \in \mathcal{N}_i} (a_{i,j} - \beta) x_j < \frac{\delta}{\alpha} \frac{1 - \epsilon}{2 - \epsilon} \frac{1}{\epsilon} + (1 - d_i \beta)(\epsilon - 1) \leq \frac{\delta}{2 \alpha \epsilon} + d_i \beta - 1 + \epsilon (1 - d_i \beta).$$

By the boundedness of $\{x_j\}_{j=1}^N$, a properly small $\delta'$ can be found. ∎

## C  SUPPLEMENTARY RESULTS AND PROOFS OF PROPOSITIONS IN SECTION 5

We assume that $a_{i,j}$ is symmetric, and $a_{i,j} > 0$ if $a_{i,j} \neq 0$. This condition means that graph is undirected. Since we deal with each channel independently, we abuse the notation to let $x_i$ denote one feature component of node $\mathbf{x}_i$ to simplifying the notation in proofs.

### C.1  THE GRAND MODEL

First, we consider the *oversmoothing* phenomenon if there is only the diffusion process with diffusion coefficients independent of $x_i$, which is a specific model of graph diffusion network (GRAND) (Chamberlain et al., 2021),

$$\dot{x}_i = \alpha \sum_{j:(i,j) \in \mathcal{E}} a_{i,j}(x_j - x_i). \tag{16}$$

**Proposition 5.** *Let $\mathbf{D}$ denote the degree matrix, i.e., $\mathbf{D} := \mathrm{diag}(d_1, \cdots, d_N)$, where $d_i = \sum_j a_{i,j}$. Then $\mathbf{D} - \mathbf{A}$ is symmetric positive semi-definite with the eigenvalues $0 = \lambda_0 \leq \lambda_1 \leq \cdots \leq \lambda_{\max} < \infty$. Let $\lambda_{\min} > 0$ be the smallest positive eigenvalue, then for all $t \geq 0$, there exists a constant $C > 0$ such that $\mathbf{E}(\mathbf{x}(t)) \leq C \exp(-\lambda_{\min}^2 t)$.*

**Proof.** Let $\mathcal{L} := \mathbf{D} - \mathbf{A}$, we have,

$$\mathbf{x}(t) = \mathbf{x}(0) e^{-\mathcal{L}}.$$

Using eigenvalue decomposition, the solution $x(t)$ writes

$$\mathbf{x}(t) = \mathbf{U}^\top e^{-\mathbf{\Lambda} t} \mathbf{U} \mathbf{x}(0) \tag{17}$$

Since the Dirichlet energy can also be written as

$$\mathbf{E}(\mathbf{x}(t)) = \mathbf{x}(t)^\top \mathcal{L} \mathbf{x}(t), \tag{18}$$

Taking (17) to (18) gives

$$\mathbf{E}(\mathbf{x}(t)) = \mathbf{x}(0)^\top \mathbf{U}^\top e^{-\mathbf{\Lambda} t} \mathbf{\Lambda} e^{-\mathbf{\Lambda} t} \mathbf{U} \mathbf{x}(0). \tag{19}$$

Therefore, $\mathbf{E}(\mathbf{x}(t)) \leq C \exp(-\lambda_{\min}^2 t)$ for some constant $C > 0$. ∎

**Proposition 6.** *We also consider a more general case,*

$$\frac{d}{dt} x_i(t) = \sum_{j:(i,j) \in \mathcal{E}} a(x_i, x_j)(x_j - x_i), \tag{20}$$

*with $a(x_i, x_j) = a(x_j, x_i) \geq a_{\min} > 0$, for any $x_i$, $x_j$.*

*Let the mass center $x_c = \frac{1}{N} \sum_{i \in \mathcal{V}} x_i$. From the symmetry of $a(x_i, x_j)$ and (20), we obtain $\mathrm{d}x_c/\mathrm{d}t = 0$ for any $t > 0$. Without loss of generality, we may assume*

$$x_c(0) = 0, \tag{21}$$

*and graph $\mathcal{G}$ is connected, i.e., $\forall (i, j) \in \mathcal{V} \times \mathcal{V}$, $\mathcal{G}$ contains a path from $i$ to $j$. Then we have, $\|x(t)\|^2 \leq \|x(0)\|^2 e^{-2a_{\min} \lambda_{\min} t}$ and $\mathbf{E}(x(t)) \leq \lambda_{\max} \|x(0)\|^2 e^{-2a_{\min} \lambda_{\min} t}$. Note that the above estimates hold true for any initial condition $x_c(0) = c$, since $x$ satisfies the ODE system (20) up to a constant. If $x_c(0) = c$, $x_i$ will converge to $c$ in time. If $\mathcal{G}$ is not connected, then we just need to consider each connected sub-graph separately with the assumption $x_{c'}(0) = \frac{1}{N'} \sum_{i \in \mathcal{V}'} x_i = c'$ for each sub-graph $\mathcal{G}' = (\mathcal{V}', \mathcal{E}')$. $x_i'$ in each sub-graph will converge to constant $c'$ independently.*

**Proof.** We multiply $x_i$ on both sides of the equation (20) and sum over $x_i$ to obtain

$$x_i \frac{dx_i}{dt} = \sum_{j \in N(j)} a(x_i, x_j)(x_j - x_i) x_i \tag{22}$$

$$\Rightarrow \quad \frac{d}{dt} \|x\|^2 = -2 \sum_{(i,j) \in \mathcal{E}} a(x_i, x_j)(x_j - x_i)^2 \tag{23}$$

$$\Rightarrow \quad \frac{d}{dt} \|x\|^2 \leqslant -2a_{\min} \sum_{(i,j) \in \mathcal{E}} (x_j - x_i)^2 \tag{24}$$

The RHS in (24) can be written in matrix form with $\mathcal{L} := \mathbf{D} - \mathbf{A}$,

$$\sum_{(i,j) \in \mathcal{E}} (x_j - x_i)^2 = \sum_{(i,j) \in \mathcal{V} \times \mathcal{V}} a_{i,j}(x_j - x_i)^2 = x^\top \mathcal{L} x.$$

Since $\mathcal{G}$ is a connected graph, $\mathbf{1}$ is the only eigenvector consisting of the kernel space of $\mathcal{L}$, therefore, $x^T \mathcal{L} x \geq \lambda_{\min} \|x\|^2$ for any $x$ satisfying $\sum_{i \in \mathcal{V}} x_i = 0$. Then, (24) leads to

$$\frac{d}{dt} \|x\|^2 \leqslant -2a_{\min} \lambda_{\min} \|x\|^2. \tag{25}$$

This yields the decay estimates for $\|x\|$ and $\mathbf{E}(x(t))$:

$$\|x(t)\|^2 \leq \|x(0)\|^2 e^{-2a_{\min}\lambda_{\min}t}, \quad \mathbf{E}(x(t)) \leq \lambda_{\max}\|x(0)\|^2 e^{-2a_{\min}\lambda_{\min}t}.$$

∎

### C.2 THE MODEL WITH ALLEN-CAHN TERM

Next, we consider the case $\beta = 0$ but with Allen-Cahn term:

$$\begin{cases} \frac{d}{dt} x_i(t) = \alpha \sum_{j:(i,j) \in \mathcal{E}} a(x_i, x_j)(x_j - x_i) + \delta x_i (1 - x_i^2), \\ a(x_i, x_j) = a(x_j, x_i) \geq 0, \quad \forall i, j \in \mathcal{V} \quad \sum_i a(x_i, x_j) = 1, \forall j \in \mathcal{V}. \end{cases} \tag{26}$$

**Proposition 7.** *Suppose* $\mathbf{x}^* = (x_1^*, \ldots, x_N^*)$ *is a global equilibrium (or steady state solution) of (26) on* $\mathbb{R}$ *and* $\mathbf{x}$*, then* $x_i^* \in [-1, 1]$.

**Proof.** Suppose $x^*$ achieves the equilibrium of (26), and $x_k^* \geq x_i^*$ $\forall i$. If $x_k^* > 1$, then $\alpha \sum_{j:(k,j) \in \mathcal{E}} a(x_k^*, x_j^*)(x_j^* - x_k^*) \leq 0$ and $x_k^*(1 - x_k^{*2}) < 0$, which contradicts with $\frac{\partial}{\partial t} x_k^* = 0$. ∎

The emergence of clusters depends on the distribution of initial features. If all the initial features are in only one potential well, then intuitively it is impossible to produce more than one cluster in the dynamics (26). As a simple transference of Lemma 3.2 in Ha et al. (2010), we can prove this. Set

$$x^M(t) := \max_i x_i(t), \quad x^m(t) := \min_i x_i(t), \tag{27}$$

where $x_i$ is still some component of node feature $\mathbf{x}_i$. Assume $x^m, x^M$ are both Lipschitz continuous and therefore they are almost differentiable everywhere in time $t$.

**Proposition 8.** *Let* $\{x_i\}$ *be the solutions of (26), then the following holds.*
*(i) If* $x^m(0) > 0$*, then* $x^m(t) \geq 0$ *for all* $t > 0$.
*(ii) If* $x^M(0) < 0$*, then* $x^M(t) \leq 0$ *for all* $t > 0$.

**Proof.** The proof was essentially given by Ha et al. (2010). For the sake of completeness, we give a proof here. (i) If $x^m(0) > 0$, we assert there exists a time sequence $\{t_j\}_{j=0}^\infty$ satisfying $t_0 = 0 < t_1 < \cdots < t_j < \ldots$, $x^m(t)$ is differentiable in each time interval $(t_{j-1}, t_j)$ and $x_i^m \geq 0$ when $t \in [0, t_1]$. By induction, firstly we set

$$x^m(t) \geq 0, \quad t \in [0, t_l].$$

If $x^m$ becomes negative in the time interval $(t_l, t_{l+1})$ there exists $t^* \in (t_l, t_{l+1})$ such that $x^m(t^*) = 0$ by the continuity of $x^m(t)$. One can assume $x_m(t) \equiv x_i(t)$ for some node $x_i$ in some time interval subset to $(t_l, t_{l+1})$. At that moment,

$$\frac{dx_i}{dt}(t^*) = \alpha \sum_j a(x_j, x_i)(x_j(t^*) - x_i(t^*)) + \delta x_i(t^*)(1 - x_i^2(t^*))$$

$$= \alpha \sum_j a(x_j, x_i)x_j(t^*) \tag{28}$$

$$\geq 0.$$

Hence, the trajectory $x^m$ becomes non-decreasing at $t = t^*$. By induction, we derive (i).

(ii) can be proved by the same argument as those for (i). ∎

Now we consider the second kinetic model (14). We can prove that if any particle $x_i$ gets caught in one potential well, then it will not escape from that well.

### C.3  THE ATTRACTIVE-REPULSIVE MODEL

We first show that the solution features of graph in Allen-Cahn model below is bounded. For simplicity of the proof, we rewrite (10) in component form where we let $a(x_i, x_j) := a_{i,j} - \beta_{i,j}$:

$$\frac{d}{dt}x_i(t) = \alpha \sum_{j:(i,j)\in\mathcal{E}} a(x_i, x_j)(x_j - x_i) + \delta x_i \left(1 - x_i^2\right). \tag{29}$$

Model (29) allows negative $a(x_i, x_j)$ which is different from the condition in (26).

**Proof of Proposition 1.**

We multiply $x_i$ on both sides of the following equation and sum over $x_i$ to obtain

$$\frac{dx_i}{dt} = \sum_{j\in\mathcal{N}_i} a(x_i, x_j)(x_j - x_i) - x_i^3 + x_i$$

$$\Rightarrow \frac{1}{2}\frac{dx_i^2}{dt} = \sum_{j\in\mathcal{N}_i} a(x_i, x_j)(x_j - x_i)x_i - x_i^4 + x_i^2 \tag{30}$$

$$\Rightarrow \frac{1}{2}\sum_{i\in\mathcal{V}}\frac{dx_i^2}{dt} = -\sum_{i\in\mathcal{V}}\left(\sum_{j\in\mathcal{N}_i} a(x_i, x_j)(x_j - x_i)x_i - x_i^4 + x_i^2\right).$$

By grouping $a(x_i, x_j)(x_j - x_i)x_i$, then

$$\frac{1}{2}\frac{d}{dt}\|x\|^2 = -\frac{1}{2}\sum_{i\in\mathcal{V}}\sum_{j\in\mathcal{N}_i} a(x_i, x_j)(x_j - x_i)^2 - \sum_{i\in\mathcal{V}} x_i^4 + \|x\|^2. \tag{31}$$

Note that $a(x_i, x_j)$ are bounded for any $(x_i, x_j)$. Let the $|a(x_i, x_j)| < D_1$ for a constant $D_1$ depending on hyper-parameters $\beta_{i,j}$. By the Cauchy-Schwarz inequality,

$$|a(x_i, x_j)(x_j - x_i)^2| \leq 2D_1(x_i^2 + x_j^2).$$

Hence,

$$-\sum_{i\in\mathcal{V}}\sum_{j\in\mathcal{N}_i} a(x_i, x_j)(x_j - x_i)^2 \leq c_4\|x\|^2.$$

Also, $\sum_{i\in\mathcal{V}} x_i^4 \geq c_3\|x\|^4$ for a constant $c_3$ depending only on $N$. Taking the above estimates to (31) gives

$$\frac{d}{dt}\|x\|^2 \leq -2c_3\|x\|^4 + (c_4 + 2)\|x\|^2.$$

If $\|x\|$ blows up for $t > 0$, the $\|x\| \to \infty$ as time $t$ increases, and $\frac{d}{dt}\|x\|^2 > 0$ for all $t$ before the blowing-up time $T_{\text{end}}$. However, one can find a $t^* < T_{\text{end}}$ such that $\|x(t^*)\|$ is large enough and

$$-2c_3\|x(t^*)\|^4 + (c_4 + 2)\|x(t^*)\|^2 < 0,$$

which produces a contradiction. Thus, $\|x\| \leq c_5$ for a constant $c_5$ only depending on $N$ and $D_1$ and

$$\mathbf{E}(x) \leq \lambda_{\max}\|x\|^2 \leq \lambda_{\max}c_5,$$

where $\lambda_{\max}$ is the largest eigenvalue of $\mathcal{L} := \mathbf{D} - \mathbf{A}$. Thus, we proved the assertion in Proposition 1.

∎

Recall (5) under (12) and rewrite it as

$$\begin{cases} \dfrac{d}{dt}x_i^{(1)} = \alpha \displaystyle\sum_{k=1}^{N_1} \overline{a}_{k,i}(x_k^{(1)} - x_i^{(1)}) - \alpha \sum_{k=1}^{N_2} \overline{a}_{k,i}(x_k^{(2)} - x_i^{(1)}) + \delta x_i^{(1)}(1 - (x_i^{(1)})^2), \ i = 1, \ldots, N_1 \\ \dfrac{d}{dt}x_j^{(2)} = \alpha \displaystyle\sum_{k=1}^{N_2} \overline{a}_{k,j}(x_k^{(2)} - x_i^{(2)}) - \alpha \sum_{k=1}^{N_1} \overline{a}_{k,j}(x_k^{(1)} - x_j^{(2)}) + \delta x_j^{(2)}(1 - (x_j^{(2)})^2), \ j = 1, \ldots, N_2. \end{cases}$$
$$(32)$$

For the attractive-repulsive model (32), we can refer to the the proof of its Theorem 5.1. in Fang et al. (2019).

We define the following notations for further proof:

$$V := \{\text{nodes indexed by } \mathcal{I}_1\}, \quad W := \{\text{nodes indexed by } \mathcal{I}_2\},$$

$$N_1 := |V|, \quad N_2 := |W|,$$

$$\widehat{x^{(1)}} := x_i^{(1)} - x_c^{(1)}, \quad \widehat{x^{(2)}} := x_i^{(2)} - x_c^{(2)},$$

$$x_c^{(1)} := \frac{1}{N_1}\sum_{i=1}^{N_1} x_i^{(1)}, \quad x_c^{(2)} := \frac{1}{N_2}\sum_{i=1}^{N_2} x_i^{(2)},$$

$$M_2(V) := \frac{1}{N_1}\sum_{i=1}^{N_1}(x_i^{(1)})^2, \quad M_2(W) := \frac{1}{N_2}\sum_{i=1}^{N_2}(x_i^{(2)})^2,$$

$$M_2 := M_2(V) + M_2(W),$$

$$\widehat{M_2} := M_2(\widehat{V}) + M_2(\widehat{W}).$$

**Remark 1.** *(13) indicates that the repulsive force between the particles should be weaker than the attractive force($S > D$).*

To prove Proposition 2, we need the following two lemmas, which we would postpone to prove.

**Lemma 1.** *Let $\{x_i\}$ be a solution to (32). Then $M_2$ satisfies*

$$\begin{aligned} \frac{d}{dt}M_2 = &-\frac{\alpha}{N_1}\sum_{i,k=1}^{N_1}\overline{a}_{k,i}(x_k^{(1)} - x_i^{(1)})^2 - \frac{2\alpha}{N_1}\sum_{k=1}^{N_2}\sum_{i=1}^{N_1}\overline{a}_{i,k}(x_k^{(2)} - x_i^{(1)})x_i^{(1)} \\ &+ \frac{2\delta}{N_1}\sum_{i=1}^{N_1}(x_i^{(1)})^2(1 - (x_i^{(1)})^2) \\ &-\frac{\alpha}{N_2}\sum_{j,k=1}^{N_2}\overline{a}_{k,j}(x_k^{(2)} - x_j^{(2)})^2 - \frac{2\alpha}{N_2}\sum_{k=1}^{N_1}\sum_{j=1}^{N_2}\overline{a}_{j,k}(x_k^{(1)} - x_j^{(2)})x_j^{(2)} \\ &+ \frac{2\delta}{N_2}\sum_{j=1}^{N_2}(x_j^{(2)})^2(1 - (x_j^{(2)})^2). \end{aligned}$$
$$(33)$$

*Suppose that the system parameters satisfy*

$$S \geq 0, \quad D > 0, \quad \delta > 0,$$

*then there exists a positive constant $M_2^\infty$ such that*

$$\sup_{0 \le t < \infty} M_2(t) \le M_2^\infty < \infty. \tag{34}$$

**Proof of Lemma 1.**

$$\frac{d}{dt} M_2(V) = \frac{2}{N_1} \sum_{i=1}^{N_1} x_i^{(1)} \dot{x}_i^{(1)}$$

$$= -\frac{\alpha}{N_1} \sum_{i,k}^{N_1} \overline{a}_{k,i} (x_k^{(1)} - x_i^{(1)})^2 - \frac{2\alpha}{N_1} \sum_{k=1}^{N_2} \sum_{i=1}^{N_1} \overline{a}_{i,k} (x_k^{(2)} - x_i^{(1)}) x_i^{(1)} \tag{35}$$

$$+ \frac{2\delta}{N_1} \sum_{i=1}^{N_1} (x_i^{(1)})^2 (1 - (x_i^{(1)})^2).$$

Similarly,

$$\frac{d}{dt} M_2(W) = \frac{2}{N_2} \sum_{i=1}^{N_2} x_i^{(2)} \dot{x}_i^{(2)}$$

$$= -\frac{\alpha}{N_2} \sum_{j,k=1}^{N_2} \overline{a}_{k,j} (x_k^{(2)} - x_j^{(2)})^2 - \frac{2\alpha}{N_2} \sum_{k=1}^{N_1} \sum_{j=1}^{N_2} \overline{a}_{j,k} (x_k^{(1)} - x_j^{(2)}) x_j^{(2)} \tag{36}$$

$$+ \frac{2\delta}{N_2} \sum_{j=1}^{N_2} (x_j^{(2)})^2 (1 - (x_j^{(2)})^2).$$

Sum the $M_2(V)$ and $M_2(W)$. Note that $\overline{a}_{ij} = \overline{a}_{ji}$. Then

$$\frac{d}{dt} M_2 \le \frac{D\alpha}{N_1} \sum_{k=1}^{N_2} \sum_{i=1}^{N_1} \left( (x_k^{(2)} - x_i^{(1)})^2 + (x_i^{(1)})^2 \right) + \frac{D\alpha}{N_2} \sum_{k=1}^{N_1} \sum_{j=1}^{N_2} \left( (x_k^{(1)} - x_j^{(2)})^2 + (x_j^{(2)})^2 \right)$$

$$+ \frac{2\delta}{N_1} \sum_{i=1}^{N_1} (x_i^{(1)})^2 (1 - (x_i^{(1)})^2) + \frac{2\delta}{N_2} \sum_{i=1}^{N_2} (x_i^{(2)})^2 (1 - (x_i^{(2)})^2). \tag{37}$$

By the Cauchy-Schwarz inequality,

$$\left( \sum_{i=1}^{N_1} (x_i^{(1)})^2 \right)^2 \le N_1 \sum_{i=1}^{N_1} (x_i^{(1)})^4, \quad \left( \sum_{i=1}^{N_1} (x_i^{(1)})^2 \right)^2 \le N_2 \sum_{i=1}^{N_2} (x_i^{(2)})^4,$$

$$(x_i^{(1)} - x_j^{(2)})^2 \le 2((x_i^{(1)})^2 + (x_j^{(2)})^2).$$

These relations and (37) yield a Riccati-type differential inequality:

$$\frac{d}{dt} M_2 \le 2D\alpha N_2 M_2(W) + 3D\alpha N_2 M_2(V) + 2D\alpha N_1 M_2(V) + 3D\alpha N_2 M_2(W)$$

$$+ 2\delta M_2 - \delta(M_2)^2 \tag{38}$$

$$\le (\alpha C_m + 2\delta) M_2 - \delta(M_2)^2.$$

Let $y$ be a solution of the following ODE:

$$y' = \alpha C_m y - \delta y^2. \tag{39}$$

Then, the solution $y(t)$ to (39) satisfies

$$M_2(t) \le y(t) \le \max \left\{ \frac{\alpha C_m}{\delta} + 2, M_2(0) \right\} =: M_2^\infty. \tag{40}$$

∎

**Lemma 2.** *Let $\{x_i\}$ be a solution to (32) with $\delta > 0$. Then $\widehat{M}_2$ satisfies*

$$\frac{d}{dt}\widehat{M}_2 \leq -2\eta\widehat{M}_2 + 2\alpha D\zeta|x_c^{(1)} - x_c^{(2)}|\sqrt{\widehat{M}_2}, \tag{41}$$

*where $\zeta = \max\{N_1, N_2\}$ and $\eta$ is the positive constant in Proposition 2.*

**Proof of Lemma 2.** By computation,

$$\dot{x}_c^{(1)} = \frac{1}{N_1}\sum_{i=1}^{N_1}\dot{x}_i^{(1)}$$

$$= \frac{\alpha}{N_1}\sum_{i,k=1}^{N_1}\bar{a}_{k,i}(x_k^{(1)} - x_i^{(1)}) - \frac{\alpha}{N_1}\sum_{k=1}^{N_2}\sum_{i=1}^{N_1}\bar{a}_{k,i}(x_k^{(2)} - x_i^{(1)}) + \frac{\delta}{N_1}\sum_{i=1}^{N_1}x_i^{(1)}(1 - (x_i^{(1)})^2)$$

$$= -\frac{\alpha}{N_1}\sum_{k=1}^{N_2}\sum_{i=1}^{N_1}\bar{a}_{k,i}(x_k^{(2)} - x_i^{(1)}) + \frac{\delta}{N_1}\sum_{i=1}^{N_1}x_i^{(1)}(1 - (x_i^{(1)})^2).$$

Note that $\dot{\widehat{x_i^{(1)}}} = \dot{x}_i^{(1)} - \dot{x}_c^{(1)}$. Take the inner product $2\widehat{x_i^{(1)}}$ with the above equation and sum it over all $i = 1, \ldots, N_1$, combining with $\sum \widehat{x_i^{(1)}} = 0$. Then,

$$\frac{d}{dt}M_2(\widehat{V}) = \frac{1}{N_1}\left[-\alpha\sum_{i,k=1}^{N_1}\bar{a}_{k,i}(\widehat{x_k^{(1)}} - \widehat{x_i^{(1)}})^2 - 2\alpha\sum_{k=1}^{N_2}\sum_{i=1}^{N_1}\bar{a}_{k,i}(x_k^{(2)} - x_i^{(1)})\widehat{x_i^{(1)}} + 2\delta\sum_{i=1}^{N_1}\widehat{x_i^{(1)}}x_i^{(1)}(1 - (x_i^{(1)})^2)\right]$$

$$= \frac{1}{N_1}\left[-\alpha\sum_{i,k=1}^{N_1}\bar{a}_{k,i}(\widehat{x_k^{(1)}} - \widehat{x_i^{(1)}})^2 - 2\alpha\sum_{i=1}^{N_1}\sum_{k=1}^{N_2}\bar{a}_{k,i}(x_c^{(2)} - x_c^{(1)} + \widehat{x_k^{(2)}} - \widehat{x_i^{(1)}})\widehat{x_i^{(1)}}\right]$$

$$+ \frac{1}{N_1}2\delta\sum_{i=1}^{N_1}\widehat{x_i^{(1)}}x_i^{(1)}(1 - (x_i^{(1)})^2).$$

Similarly,

$$\frac{d}{dt}M_2(\widehat{W}) = \frac{1}{N_2}\left[-\alpha\sum_{i,k=1}^{N_2}\bar{a}_{k,i}(\widehat{x_k^{(2)}} - \widehat{x_i^{(2)}})^2 - 2\alpha\sum_{k=1}^{N_1}\sum_{j=1}^{N_2}\bar{a}_{k,j}(x_c^{(1)} - x_c^{(2)} + \widehat{x_k^{(1)}} - \widehat{x_j^{(2)}})\widehat{x_j^{(2)}}\right]$$

$$+ \frac{1}{N_2}2\delta\sum_{i=1}^{N_2}\widehat{x_i^{(2)}}x_i^{(2)}(1 - (x_i^{(2)})^2).$$

Combine the two equations, $\frac{d}{dt}\widehat{M}_2 = \sum_{i=1}^{6}I_i$, where

$$I_1 := \frac{1}{N_1}\left[-\alpha\sum_{i,k=1}^{N_1}\bar{a}_{k,i}(\widehat{x_k^{(1)}} - \widehat{x_i^{(1)}})^2\right] \leq -2\alpha SN_1 M_2(\widehat{V}),$$

$$I_2 := \frac{1}{N_2}\left[-\alpha\sum_{i,k=1}^{N_2}\bar{a}_{k,i}(\widehat{x_k^{(2)}} - \widehat{x_i^{(2)}})^2\right] \leq -2\alpha SN_2 M_2(\widehat{W}), \tag{42}$$

$$I_1 + I_2 \leq -\alpha S\min\{N_1 N_2\}\widehat{M}_2,$$

$$I_3 := -2\alpha \sum_{k=1}^{N_2} \sum_{i=1}^{N_1} \overline{a}_{k,i} \widehat{(x_k^{(2)} - x_i^{(1)})} \widehat{x_i^{(1)}} \frac{1}{N_1} - 2\alpha \sum_{k=1}^{N_1} \sum_{j=1}^{N_2} \overline{a}_{j,k} \widehat{(x_k^{(1)} - x_j^{(2)})} \widehat{x_j^{(2)}} \frac{1}{N_2}$$

$$\leq \max\left\{ \frac{1}{N_1}, \frac{1}{N_2} \right\} 2\alpha \sum_{i=1}^{N_1} \sum_{j=1}^{N_2} \overline{a}_{i,j} \widehat{(x_i^{(1)} - x_j^{(2)})}^2$$

$$\leq \max\left\{ \frac{1}{N_1}, \frac{1}{N_2} \right\} 2\alpha D \sum_{i=1}^{N_1} \sum_{j=1}^{N_2} \widehat{(x_i^{(1)} - x_j^{(2)})}^2$$

$$= 2\alpha D \max\left\{ \frac{1}{N_1}, \frac{1}{N_2} \right\} N_1 N_2 \widehat{M_2}$$

$$= 2\alpha D \zeta \widehat{M_2},$$

(43)

$$I_4 := -2\alpha \sum_{k=1}^{N_2} \sum_{i=1}^{N_1} \overline{a}_{k,i} (x_c^{(2)} - x_c^{(1)}) \widehat{x_i^{(1)}} \frac{1}{N_1} - 2\alpha \sum_{k=1}^{N_1} \sum_{j=1}^{N_2} \overline{a}_{j,k} (x_c^{(1)} - x_c^{(2)}) \widehat{x_j^{(2)}} \frac{1}{N_2}$$

$$\leq 2\alpha D \zeta |x_c^{(1)} - x_c^{(2)}| \sqrt{\widehat{M_2}},$$

$$I_5 := 2\delta \sum_{i=1}^{N_1} \widehat{x_i^{(1)}} x_i^{(1)} (1 - (x_i^{(1)})^2) \frac{1}{N_1},$$

$$I_6 := 2\delta \sum_{i=1}^{N_2} \widehat{x_i^{(2)}} x_i^{(2)} (1 - (x_i^{(2)})^2) \frac{1}{N_2}.$$

(44)

Using $x_i^{(1)} = \widehat{x_i^{(1)}} + x_c^{(1)}$ and $\sum_i \widehat{x_i^{(1)}} = 0$, we obtain

$$I_5 = \frac{2\delta}{N_1} \sum_{i=1}^{N_1} (1 - (x_i^{(1)})^2) \widehat{x_i^{(1)}}^2 + \frac{2\delta}{N_1} \sum_{i=1}^{N_1} x_c^{(1)} \widehat{x_i^{(1)}}$$

$$= 2\delta M_2(\widehat{V}) - \frac{2\delta}{N_1} \sum_{i=1}^{N_1} (x_i^{(1)})^2 \widehat{x_i^{(1)}}^2 - \frac{2\delta}{N_1} \sum_{i=1}^{N_1} (x_i^{(1)})^2 x_c^{(1)} \widehat{x_i^{(1)}}$$

$$= 2\delta M_2(\widehat{V}) - \frac{2\delta}{N_1} \sum_{i=1}^{N_1} (x_i^{(1)})^3 \widehat{x_i^{(1)}}$$

$$\leq 2\delta M_2(\widehat{V}).$$

(45)

The last inequality is based on

$$\sum_{i=1}^{N_1} (x_i^{(1)})^3 \widehat{x_i^{(1)}} = \sum_{i=1}^{N_1} (x_i^{(1)})^2 ((x_i^{(1)})^2 - x_c^{(1)} \widehat{x_i^{(1)}})$$

$$= \frac{1}{2} \sum_{i=1}^{N_1} (x_i^{(1)})^2 ((x_i^{(1)})^2 - (x_c^{(1)})^2 + (x_i^{(1)} - x_c^{(1)})^2)$$

$$\geq \frac{1}{2} \sum_{i=1}^{N_1} (x_i^{(1)})^2 ((x_i^{(1)})^2 - (x_c^{(1)})^2)$$

$$= \frac{1}{2} \sum_{i=1}^{N_1} (x_i^{(1)})^4 - \frac{1}{2} \sum_{i=1}^{N_1} (x_i^{(1)})^2 ((x_c^{(1)})^2$$

$$\geq \frac{1}{2} \sum_{i=1}^{N_1} (x_i^{(1)})^4 - \frac{1}{2N_1} \left( \sum_{i=1}^{N_1} (x_i^{(1)})^2 \right)^2 \geq 0.$$

Similarly on $I_6$, one has $I_6 \leq 2\delta M_2(\widehat{W})$. Thus, $I_5 + I_6 \leq 2\delta \widehat{M_2}$.

Note that

$$
\begin{aligned}
I_1 + I_2 + I_3 + I_5 + I_6 &\leq -2\alpha S \min\{N_1, N_2\}\widehat{M_2} + 2\alpha D\zeta \widehat{M_2} + 2\delta \widehat{M_2} \\
&\leq -2\left[\alpha(S-D)\min\{N_1, N_2\} - \delta\right]\widehat{M_2} \\
&\leq -2\eta \widehat{M_2}.
\end{aligned}
\tag{46}
$$

∎

**Proof of Proposition 2**

(a) (Uniform upper bound of $|x_c^{(1)} - x_c^{(2)}|$) By Cauchy's inequality and Lemma 1,

$$
\begin{aligned}
|x_c^{(1)} - x_c^{(2)}| &= \left| \frac{1}{N_1} \sum_{i=1}^{N_1} x_i^{(1)} - \frac{1}{N_2} \sum_{i=1}^{N_2} x_i^{(2)} \right| \\
&\leq \frac{1}{N_1} \sum_{i=1}^{N_1} |x_i^{(1)}| + \frac{1}{N_2} \sum_{i=1}^{N_2} |x_i^{(2)}| \\
&\leq 2\sqrt{\frac{1}{N_1} \sum_{i=1}^{N_1} (x_i^{(1)})^2 + \frac{1}{N_2} \sum_{i=1}^{N_2} (x_i^{(2)})^2} \\
&= 2\sqrt{M_2(t)} \leq 2\sqrt{M_2^\infty}.
\end{aligned}
\tag{47}
$$

(b) (Uniform boundedness of $\widehat{M_2}$) By Lemma 2 and (47),

$$
\begin{aligned}
\frac{d}{dt}\sqrt{\widehat{M_2}} &\leq -\eta\sqrt{\widehat{M_2}} + \alpha D\zeta |x_c^{(1)} - x_c^{(2)}| \\
&\leq -\eta\sqrt{\widehat{M_2}} + 2D\alpha\zeta\sqrt{M_2^\infty}.
\end{aligned}
\tag{48}
$$

Use Gronwall's lemma to obtain

$$
\begin{aligned}
\sqrt{\widehat{M_2}(t)} &\leq \sqrt{\widehat{M_2}(0)}e^{-\eta t} + \frac{2D\alpha\zeta\sqrt{M_2^\infty}}{\eta}(1 - e^{-\eta t}) \\
&\leq \max\left\{ \sqrt{\widehat{M_2}(0)}, \frac{2D\alpha\zeta\sqrt{M_2^\infty}}{\eta} \right\} := C_3.
\end{aligned}
\tag{49}
$$

(c) (Separation of the particle centers)

By (32), we have

$$
\begin{aligned}
\frac{d}{dt}|x_c^{(1)} - x_c^{(2)}| =\ & \frac{\alpha}{N_1} \sum_{i=1}^{N_1}\sum_{k=1}^{N_2} \overline{a}_{k,i}(x_k^{(1)} - x_i^{(1)}) - \frac{\alpha}{N_2} \sum_{j=1}^{N_2}\sum_{k=1}^{N_1} \overline{a}_{k,j}(x_k^{(2)} - x_j^{(2)}) \\
& - \frac{\alpha}{N_1} \sum_{i=1}^{N_1}\sum_{k=1}^{N_2} \overline{a}_{k,i}(x_k^{(2)} - x_i^{(1)}) - \frac{\alpha}{N_2} \sum_{j=1}^{N_2}\sum_{k=1}^{N_1} \overline{a}_{k,i}(x_k^{(1)} - x_j^{(2)}) \\
& + \frac{\delta}{N_1} \sum_{i=1}^{N_1} (x_i^{(1)}(1 - ((x_i^{(1)})^2) - \frac{2\delta}{N_2} \sum_{i=1}^{N_2}(x_i^{(2)}(1 - ((x_i^{(2)})^2)
\end{aligned}
\tag{50}
$$

By the symmetry of $(\overline{a}_{i,j})$, $\sum_{i=1}^{N_p} \sum_{k=1}^{N_p} \overline{a}_{k,i}(\widehat{x_k^{(p)}} - \widehat{x_i^{(p)}}) = 0$ for $p = 1, 2$. Thus,

$$
\begin{aligned}
\frac{d}{dt}|x_c^{(1)} - x_c^{(2)}| =& \frac{\alpha}{N_1} \sum_{i=1}^{N_1} \sum_{k=1}^{N_2} \overline{a}_{k,i}(\widehat{x_k^{(2)}} + x_c^{(2)} - \widehat{x_i^{(1)}} - x_c^{(1)}) \\
&+ \frac{\alpha}{N_2} \sum_{j=1}^{N_2} \sum_{k=1}^{N_1} \overline{a}_{k,j}(\widehat{x_k^{(1)}} + x_c^{(1)} - \widehat{x_i^{(2)}} - x_c^{(2)}) \\
&+ \frac{\delta}{N_1} \sum_{i=1}^{N_1} (\widehat{x_i^{(1)}} + x_c^{(1)}) - \frac{2\delta}{N_2} \sum_{i=1}^{N_2} (\widehat{x_i^{(2)}} + x_c^{(2)}) \\
&- \frac{2\delta}{N_1} \sum_{i=1}^{N_1} (x_i^{(1)})^3 + \frac{2\delta}{N_2} \sum_{i=1}^{N_2} (x_i^{(2)})^3
\end{aligned}
\tag{51}
$$

By a similar estimate with Lemma 2, we have

$$
\begin{aligned}
\frac{d}{dt}|x_c^{(1)} - x_c^{(2)}|^2 =& -2(x_c^{(1)} - x_c^{(2)})\frac{\alpha}{N_1} \sum_{i=1}^{N_1} \sum_{k=1}^{N_2} \overline{a}_{k,i}(\widehat{x_k^{(2)}} + x_c^{(2)} - \widehat{x_i^{(1)}} - x_c^{(1)}) \\
&+ 2(x_c^{(1)} - x_c^{(2)})\frac{\alpha}{N_2} \sum_{j=1}^{N_2} \sum_{k=1}^{N_1} \overline{a}_{k,j}(\widehat{x_k^{(1)}} + x_c^{(1)} - \widehat{x_i^{(2)}} - x_c^{(2)}) \\
&+ \left( \frac{2\delta}{N_1} \sum_{i=1}^{N_1} \widehat{x_i^{(1)}} - \frac{2\delta}{N_2} \sum_{i=1}^{N_2} \widehat{x_i^{(2)}} \right)(x_c^{(1)} - x_c^{(2)}) \\
&+ \left( \frac{2\delta}{N_1} \sum_{i=1}^{N_1} x_c^{(1)} - \frac{2\delta}{N_2} \sum_{i=1}^{N_2} x_c^{(2)} \right)(x_c^{(1)} - x_c^{(2)}) \\
&- \frac{2\delta}{N_1} \sum_{i=1}^{N_1} (x_i^{(1)})^3 (x_c^{(1)} - x_c^{(2)}) + \frac{2\delta}{N_2} \sum_{i=1}^{N_2} (x_i^{(2)})^3 (x_c^{(1)} - x_c^{(2)}) \\
=& \frac{2\alpha}{N_1} \sum_{i=1}^{N_1} \sum_{k=1}^{N_2} \overline{a}_{k,i}(x_c^{(2)} - x_c^{(1)})^2 + \frac{2\alpha}{N_1} \sum_{j=1}^{N_2} \sum_{k=1}^{N_1} \overline{a}_{k,i}(x_c^{(2)} - x_c^{(1)})^2 \\
&+ \frac{2\alpha}{N_1} \sum_{i=1}^{N_1} \sum_{k=1}^{N_2} \overline{a}_{k,i}(\widehat{x_k^{(2)}} - \widehat{x_i^{(1)}})(x_c^{(2)} - x_c^{(1)}) + \frac{2\alpha}{N_1} \sum_{j=1}^{N_2} \sum_{k=1}^{N_1} \overline{a}_{k,i}(\widehat{x_k^{(2)}} - \widehat{x_i^{(1)}})(x_c^{(2)} - x_c^{(1)}) \\
&+ 2(x_c^{(2)} - x_c^{(1)}) \left( \frac{\delta}{N_1} \sum_{i=1}^{N_1} \widehat{x_i^{(1)}} - \frac{\delta}{N_2} \sum_{j=1}^{N_2} \widehat{x_j^{(2)}} \right) \\
&+ 2(x_c^{(2)} - x_c^{(1)}) \left( \frac{\delta}{N_1} \sum_{i=1}^{N_1} x_c^{(1)} - \frac{\delta}{N_2} \sum_{j=1}^{N_2} x_c^{(2)} \right).
\end{aligned}
\tag{52}
$$

$$
\frac{d}{dt}|x_c^{(1)} - x_c^{(2)}|^2 \geq 2 \left( \alpha \left( \frac{1}{N_1} + \frac{1}{N_2} \right) \sum_{i=1}^{N_1} \sum_{j=1}^{N_2} \overline{a}_{i,j} + 2\delta \right)(x_c^{(1)} - x_c^{(2)})^2 + \mathcal{I}_{c1} + \mathcal{I}_{c2}. \tag{53}
$$

where

$$
\mathcal{I}_{c1} := 2\alpha \left( \frac{1}{N_1} + \frac{1}{N_2} \right) \sum_{i=1}^{N_1} \sum_{j=1}^{N_2} \overline{a}_{i,j}(\widehat{x_j^{(2)}} - \widehat{x_i^{(1)}})(x_c^{(2)} - x_c^{(1)}), \tag{54}
$$

$$\mathcal{I}_{c2} := -\frac{2\delta}{N_1}\sum_{i=1}^{N_1}(x_i^{(1)})^3(x_c^{(1)} - x_c^{(2)}) + \frac{2\delta}{N_2}\sum_{i=1}^{N_2}(x_i^{(2)})^3(x_c^{(1)} - x_c^{(2)}). \tag{55}$$

By the Cauchy-Schwarz inequality,

$$|\mathcal{I}_{c1}| \le 2\alpha\left(\frac{1}{N_1} + \frac{1}{N_2}\right)D\sqrt{N_1 N_2}|x_c^{(1)} - x_c^{(2)}|\sqrt{\sum_{i,j}^{N_1,N_2}(\widehat{x_j^{(2)}} - \widehat{x_i^{(1)}})^2}$$

$$\le 2\alpha\left(\frac{1}{N_1} + \frac{1}{N_2}\right)DN_1 N_2|x_c^{(1)} - x_c^{(2)}|\sqrt{\widehat{M_2}}. \tag{56}$$

For $\mathcal{I}_{c2}$, note that

$$\left|\frac{2\delta}{N_1}\sum_{i=1}^{N_1}(x_i^{(1)})^3\right| \le \delta|x_i^{(1)}|M_2(V) \le \delta\sqrt{N_1}M_2(V)^{\frac{3}{2}},$$

$$\left|\frac{2\delta}{N_2}\sum_{i=1}^{N_2}(x_i^{(2)})^3\right| \le \delta|x_i^{(2)}|M_2(V) \le \delta\sqrt{N_1}M_2(W)^{\frac{3}{2}}.$$

Then, one gets

$$\mathcal{I}_{c2} \ge -2\left|x_c^{(1)} - x_c^{(2)}\right|\left|\frac{\delta}{N_1}\sum_{i=1}^{N_1}(x_i^{(1)})^3 + \frac{\delta}{N_2}\sum_{i=1}^{N_2}(x_i^{(2)})^3\right|$$

$$\ge -2\left|x_c^{(1)} - x_c^{(2)}\right|\delta\sqrt{\max\{N_1, N_2\}}M_2(t)^{\frac{3}{2}}. \tag{57}$$

Hence,

$$\frac{d}{dt}|x_c^{(1)} - x_c^{(2)}|^2 \ge \left[\left(2\alpha(\frac{1}{N_1} + \frac{1}{N_2})\sum_{i=1}^{N_1}\sum_{j=1}^{N_2}\bar{a}_{i,j}\right) + 4\delta\right]|x_c^{(1)} - x_c^{(2)}|^2$$

$$- 2\alpha D(N_1 + N_2)\left|x_c^{(1)} - x_c^{(2)}\right|\sqrt{\widehat{M_2}} - 2\delta\sqrt{\max\{N_1, N_2\}}\left|x_c^{(1)} - x_c^{(2)}\right|M_2^{\frac{3}{2}}. \tag{58}$$

Combining with Lemma 1 and (49), one obtains the estimate

$$\frac{d}{dt}|x_c^{(1)} - x_c^{(2)}| \ge \left(\alpha\left(\frac{1}{N_1} + \frac{1}{N_2}\right)\sum_i^{N_1}\sum_{j=1}^{N_2}\bar{a}_{i,j} + 2\delta\right)|x_c^{(1)} - x_c^{(2)}|$$

$$- \alpha D(N_1 + N_2)C_3 - \delta\sqrt{\max\{N_1, N_2\}}(M_2^\infty)^{\frac{3}{2}}. \tag{59}$$

By Gronwall's lemma, if the initial data satisfy:

$$|x_c^{(1)}(0) - x_c^{(2)}(0)| \ge \frac{\alpha D(N_1 + N_2)C_3 + \delta\sqrt{\max\{N_1, N_2\}}(M_2^\infty)^{\frac{3}{2}}}{2\delta} := \frac{C_4}{\delta}, \tag{60}$$

then,

$$|x_c^{(1)}(t) - x_c^{(2)}(t)| \ge \frac{C_4}{\delta} + (|x_c^{(1)}(0) - x_c^{(2)}(0)| - \frac{C_4}{\delta})e^{\delta t} \ge \frac{C_4}{\delta}. \tag{61}$$

(d)(Spatial separation of the two sub-ensembles) For any $i = 1, \ldots N_1, j = 1, \ldots, N_2$,

$$|x_i^{(1)}(t) - x_j^{(1)}(t)| \ge |x_c^{(1)}(t) - x_c^{(2)}(t)| - |\widehat{x^{(1)}}_i(t) - \widehat{x^{(2)}}_j(t)|$$

$$\ge |x_c^{(1)}(t) - x_c^{(2)}(t)| - \sqrt{2\max\{N_1, N_2\}\widehat{M_2}}$$

$$\ge \frac{C_4}{\delta} + \left(|x_c^{(1)}(0) - x_c^{(2)}(0)| - \frac{C_4}{\delta}\right)e^{\delta t}$$

$$- \sqrt{2\max\{N_1, N_2\}}\left(\sqrt{\widehat{M_2}(0)}e^{-\eta t} + \frac{\sqrt{M_2^\infty}}{\eta}(1 - e^{-\eta t})\right).$$

Then, there exists some time $T^*$ such that $\forall t \geq T^*$,

$$|x_i^{(1)}(t) - x_j^{(2)}(t)| \geq C' > 0, \quad \forall i, j. \tag{62}$$

Combing with Proposition 1, we finish the proof. ∎

**Remark 2.** *The proof of Proposition 3 is included in part (d) of proof of Proposition 2.*

Now denote $\eta_2 := \sum_{i \in \mathcal{I}_1, j \in \mathcal{I}_2} a_{i,j} > 0$ in some channel, then the Dirichlet energy in this channel has a lower bound:

$$
\begin{aligned}
\mathbf{E}(x) &= \frac{1}{N} \sum_{i,j} a_{i,j}(x_i - x_j)^2 \\
&= \frac{1}{N} \left[ \sum_{i,j \in \mathcal{I}_1} a_{i,j}(x_i^{(1)} - x_j^{(1)})^2 + \sum_{i,j \in \mathcal{I}_2} a_{i,j}(x_i^{(2)} - x_j^{(2)})^2 + \sum_{i \in \mathcal{I}_1, j \in \mathcal{I}_2} a_{i,j}(x_i^{(1)} - x_j^{(2)})^2 \right] \\
&\geq \frac{1}{N} \sum_{i \in \mathcal{I}_1, j \in \mathcal{I}_2} a_{i,j}(x_i^{(1)} - x_j^{(2)})^2 \\
&\geq \frac{C^2 \eta_2}{N}.
\end{aligned}
\tag{63}
$$

# D    EXPERIMENTS

The code for the experiments is available at:

https://github.com/ykiiiiii/ACMP

We will replace this anonymous link with a non-anonymous GitHub link after the acceptance. We implement all experiments in Python 3.8.13 with PyTorch Geometric on one NVIDIA ® Tesla A100 GPU with 6,912 CUDA cores and 80GB HBM2 mounted on an HPC cluster.

In addition, we take the official implementation of the Graph Neural Diffusion (GRAND) as diffusion term in (9) from the repository:

https://github.com/twitter-research/graph-neural-pde

## D.1    DETAILS FOR EXPERIMENTS

**Datasets**    We consider two types of datasets: Homophilic and Heterophilic. They are differentiated by the *homophily level* of a graph (Pei et al., 2020):

$$\mathcal{H} = \frac{1}{|V|} \sum_{v \in V} \frac{\text{Number of } v\text{'s neighbors who have the same label as } v}{\text{Number of } v\text{'s neighbors}}.$$

In the experiments, we have used six homophilic datasets, including Cora (McCallum et al., 2000), Citeseer (Sen et al., 2008) and Pubmed (Namata et al., 2012), Computer and Photo (Namata et al., 2012), and CoauthorCS (Shchur et al., 2018), and three heterophilic datasets: Cornell, Texas and Wisconsin from the WebKB dataset[2]. For completeness, we list the numbers of classes, features, nodes and edges of each dataset, and their homophily level in Table 3. The low homophily level means that the dataset is more heterophilic when most of neighbours are not in the same class, and the high homophily level indicates that the dataset close to homophilic when similar nodes tent to be connected. The datasets we used in Table 3 covers various homophily levels.

**Experiment setup**    For homophilic datasets, we use 10 random weight initializations and 100 random splits, which contains 1,000 tests. Each combination randomly select 20 numbers for each class. For heterophilic data, we use the original fixed 10 split datasets. We fine-tune our model

---

[2] http://www.cs.cmu.edu/afs/cs.cmu.edu/project/theo-11/www/wwkb/

Table 3: Information for Graph Datasets Used in Experiments

| Dataset | Classes | Features | #Nodes | Edges | Homophily level |
|---|---|---|---|---|---|
| Cora | 7 | 1433 | 2485 | 5069 | 0.83 |
| CiteSeer | 6 | 3703 | 2120 | 3679 | 0.71 |
| PubMed | 3 | 500 | 19717 | 44324 | 0.79 |
| CoauthorCS | 15 | 6805 | 18333 | 81894 | 0.80 |
| Computer | 10 | 767 | 13381 | 245778 | 0.77 |
| Photo | 8 | 745 | 7487 | 119043 | 0.83 |
| Texas | 5 | 1703 | 183 | 309 | 0.11 |
| Wisconsin | 5 | 1703 | 183 | 499 | 0.21 |
| Cornell | 5 | 1703 | 183 | 499 | 0.30 |

within hyper-parameter search space, as detailed in Table 4. We use the Dormand–Prince adaptive step size scheme (DOPRI5) as the neural ODE solver for all datasets. Hyperparameter search used Ray Tune with a hundred trials using an asynchronous hyperband scheduler with a grace period of 50 epochs. All the details to reproduce our results have been included in the submission and will be publicly available after publication.

Table 4: Hyperparameter Search Space

| Hyperparameters | Search Space | Distribution |
|---|---|---|
| learning rate | $[10^{-6}, 10^{-1}]$ | log-uniform |
| weight decay | $[10^{-3}, 10^{-1}]$ | log-uniform |
| dropout rate | $[0.1, 0.8]$ | uniform |
| hidden dim | $\{64, 128, 256\}$ | categorical |
| time (T) | $[2, 25]$ | uniform |
| $\beta$ | $[0, 1]$ | uniform |

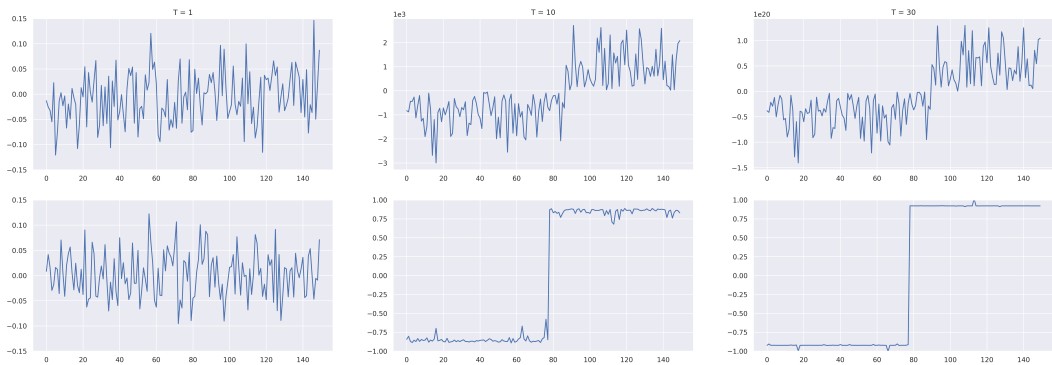

Figure 5: Example of how adding Allen-Cahn terms can prevent the nodes feature from becoming infinite. We choose the first channel in the node's feature of dimension 150. In the first row, the repulsive force is added to message passing without Allen-Cahn term, and in the second row, Allen-Cahn term is added to message passing. The first, second and third columns show the neural ODE's initial state, and the states when $T = 10$ and $T = 30$.

## D.2 ABLATION STUDY FOR ACMP

**Message Passing Performance vs Depths** We compare ACMP with various GNN models such as GRAND, GCN, GAT, and GraphSage with different depths on the planetoid datasets. Table 5

lists the nodes classification accuracy on Cora, Citeseer and Pubmed. We observe that ACMP can maintain its model performance as the network deepens and achieve top test accuracy among all listed models using the same depth. ACMP can thus overcome the oversmoothing.

Table 5: Test Accuracy of Models with Different Depth

| Model | depth | Cora | CiteSeer | PubMed |
|---|---|---|---|---|
| GRAND-1 | 4 | $82.80 \pm 1.62$ | $73.87 \pm 2.12$ | $78.71 \pm 1.19$ |
| | 16 | $82.75 \pm 1.17$ | $72.61 \pm 2.42$ | $78.79 \pm 0.93$ |
| | 32 | $82.19 \pm 1.73$ | $72.65 \pm 3.15$ | $78.70 \pm 1.08$ |
| | 64 | $80.87 \pm 2.28$ | $\mathbf{69.84} \pm 2.66$ | NA |
| | 128 | $77.22 \pm 2.88$ | NA | NA |
| GCN | 4 | $81.35 \pm 1.27$ | $70.54 \pm 6.61$ | $77.15 \pm 3.00$ |
| | 16 | $19.70 \pm 7.06$ | $24.78 \pm 1.45$ | $41.36 \pm 1.77$ |
| | 32 | $21.86 \pm 6.09$ | $24.23 \pm 1.65$ | $40.66 \pm 1.86$ |
| GAT | 4 | $80.95 \pm 2.28$ | $72.31 \pm 2.82$ | $77.37 \pm 1.32$ |
| | 16 | $29.14 \pm 1.02$ | $24.84 \pm 1.45$ | $39.21 \pm 0.43$ |
| | 32 | $29.75 \pm 1.57$ | $24.83 \pm 1.45$ | $39.02 \pm 0.12$ |
| GraphSage | 4 | $79.83 \pm 2.43$ | $50.00 \pm 14.27$ | $76.01 \pm 2.35$ |
| | 16 | $25.52 \pm 6.45$ | $24.84 \pm 1.45$ | $37.55 \pm 3.92$ |
| | 32 | $29.14 \pm 1.02$ | $28.38 \pm 2.54$ | $39.21 \pm 4.39$ |
| ACMP (ours) | 4 | $\mathbf{83.87} \pm 0.52$ | $\mathbf{74.61} \pm 1.04$ | $\mathbf{79.74} \pm 0.24$ |
| | 16 | $\mathbf{83.19} \pm 0.67$ | $\mathbf{73.13} \pm 0.85$ | $\mathbf{79.16} \pm 0.36$ |
| | 32 | $\mathbf{83.11} \pm 0.81$ | $\mathbf{72.76} \pm 1.05$ | $\mathbf{79.81} \pm 1.61$ |
| | 64 | $\mathbf{80.48} \pm 1.21$ | $68.92 \pm 1.37$ | $\mathbf{78.01} \pm 0.01$ |
| | 128 | $\mathbf{80.30} \pm 1.18$ | $\mathbf{67.83} \pm 0.02$ | $\mathbf{77.98} \pm 0.01$ |

Table 6: The number of parameters for different models

| | GCN | GAT | GraphSage | CGNN | GRAND-l | ACMP_GCN | ACMP_GAT |
|---|---|---|---|---|---|---|---|
| Cora | 144k | 230k | 200k | 26k | 17k | 17k | 19k |

**Model parameter comparison**  We compare the number of parameters of our model with different benchmark model on Cora dataset in Table 6. The depth of all model is chosen as the number which achieves the best performance on Cora dataset. We show that ACMP is a light-weight neural network architecture which can achieve good classification performance with fewer parameters.

**Allen-Cahn term**  We now show in Figure 5 how Allen-Cahn term can stabilize training and prevent node features from blowing up. The first row is the evolution of the diffusion equation without Allen-Cahn term while the second row has Allen-Cahn term added. We can observe that introducing the repulsive term is essential for bounding GNN outputs, particularly when learning heterophilic datasets. However, naively adding $\beta$ to message passing will result in all node's features becoming infinite. In the first row of Figure 5 when Allen-Cahn term is not incorporated, the node's features have increased to $3 \times 10^3$ when $T = 10$, from $0.1$ when $T = 1$. By the time $T$ equals 30, the node's largest feature becomes $1 \times 10^{20}$, which the neural ODE solver and message passing can hardly handle numerically corrected. When we introduce Allen-Cahn term, the system contains two strong attractors of $\pm 1$, and the nodes are attracted to the two ends of $1$ and $-1$ by their own features.

