# OpenReview forum: "ACMP: Allen-Cahn Message Passing with Attractive and Repulsive Forces for Graph Neural Networks"
_ICLR.cc/2023/Conference — ICLR 2023 notable top 25%_

### Official Review · Reviewer_boSt · 2022-10-23

**Confidence:** 4
**Correctness:** 3
**Technical Novelty And Significance:** 3
**Empirical Novelty And Significance:** 3
**Recommendation:** 6

**Clarity, Quality, Novelty And Reproducibility:**

The paper is well-organized. There are enough experimental details and hyperparameters. Although the source code is not provided, I still think that the purposed method of this paper can be reproduced. The authors propose that Allen-Cahn can reduce the oversmoothing phenomenon by keeping the Dirichlet energy of GNN. Allen-Cahn itself is not the way proposed by the author. So there is some novelty in this paper, but not very significant

**Strength And Weaknesses:**

Strength
1.	Using the property of ACMP to help keep the Dirichlet energy and help solve the oversmoothing of GNN is a good attempt.
2.	The experimental results show that ACMP can improve the performance of existing GNN models in the node classification task.
3.	In this paper, the mathematical arguments are detailed without obvious errors.

Weakness
1.	As this ACMP is focused on the oversmoothing problem of GNN. It should work for more tasks. The author only tests it at the node classification task. This cannot show that the purposed method can help solve the oversmoothing of all kinds of GNN task.
2.	The purposed method is not the first method to try to solve this problem. So, there should be experiments to show this method is better than the former methods.


**Summary Of The Paper:**

Firstly, in this paper, the authors purpose that ACMP can help solve the over-smoothing problem in the GNN network by keeping the Dirichlet energy of GNN. Then, this paper shows that adding the term repulsive force may cause the particles to be pushed away to infinite, the Dirichlet energy becomes unbounded, and how the purposed method helps solve the problem. Finally, the author uses the experimental results to show that ACMP can keep Dirichlet energy in GNN and help improve the performance of GNN methods.

### Update after reading rebuttal
The authors address my concerns and I raised my score.

**Summary Of The Review:**

The purposed method can help solve the GNN oversmoothing phenomenon by keeping the Dirichlet energy. The experiment result shows this. However, this paper lacks the experiment to compare the performance of the purposed method and the former method that overcomes GNN oversmoothing. This paper uses the properties of Allen-Cahn. Therefore, the novelty of this paper is not very significant.

---

> ### Author Response · Authors · 2022-11-16
> **Response to Reviewer boSt**
>
> Thank you for your review. We address your concerns below in two main parts: novelty and experiments.
>
> **About novelty:**
>
> We think our model has contributed a novel GNN model different from existing message passings. ACMP is an attempt to combine interacting particle system theory with GNN message passing. Under this formulation, we design ACMP with a clear dynamical structure and give a detailed analysis of the lower bound of its Dirichlet energy.
>
> ACMP introduced two main new terms for the equation governing the message passing: 1. attraction and repulsion for separability of node features (limiting oversmoothing); and 2. the double-well potential for trapping features, as illustrated in Fig. 2. The trade-off of the two terms is the key mechanism which makes the model succeed. Note the ACMP borrows the name from the Allen-Cahn potential (or the double-well potential), but the dynamics behind ACMP are in fact not as the Allen-Cahn equation evolution. The energy that drives ACMP’s equation is pseudo-Ginzburg-Landau energy, which enriches message passing dynamics and enables ACMP-based GNN to have an excellent performance on datasets with very different homophily levels.
>
> On the other hand, the Allen-Cahn term $\delta x(1-x^2),$ is used here to limit the whole system from blowing up. We choose the double-well potential as a simple but efficient potential form. Though the double-well potential has some similarities with the classical Allen-Cahn model (it is why we named our model ACMP), to our knowledge, it is a very new GNN model. We need to say that we have other candidates other than the double well potential for our framework, such as multi-well potential stemming from a polynomial with a certain/higher order as mentioned in Appendix B. Using multi-well potential for example will not affect the separability of the network or energy blow-up suppression but may lead to slowing the training speed.
>
>
> **About experiments:**
>
> We are grateful for your pointing out that our model should solve oversmoothing at both node and graph levels.  For graph-level tasks, we add both protein and molecule tasks, including PPI and ZINC. ACMP can be interpreted as a feature extraction that provides different modes of each graph or molecule.
>
> Protein-Protein Interactions (PPI):
> | model             | GCN  | GAT  | GraphSAGE | PDE-GCN | GCNII | GeniePath | JKNet | ACMP_GCN |
> |:-------------------:|:------:|:------:|:-----------:|:---------:|:-------:|:-----------:|:-------:|:----------:|
> | Micro-averaged F1 | 98.5 | 97.3 | 61.2      | 99.2    | 99.5  | 98.5      | 97.6  | 99.3     |
>
> Molecular graph property regression: ZINC (small 12k version)
> | model | GCN            | GAT            | GIN            | GatedGCN       | GraphSAGE      | PNA            | ACMP_GCN       |
> |:-------:|:----------------:|:----------------:|:----------------:|:----------------:|:----------------:|:----------------:|:----------------:|
> | MAE   | 0.47$\pm$0.002 | 0.46$\pm$0.002 | 0.41$\pm$0.008 | 0.42$\pm$0.006 | 0.41$\pm$0.005 | 0.32$\pm$0.032 | 0.23$\pm$0.008 |
>
> *"The proposed method is not the first method to try to solve this problem. So, there should be experiments to show this method is better than the former methods."*
>
> Besides analyzing the lower and upper bounds of the Dirichlet energy, we also conduct several experiments to demonstrate that our model is not affected by oversmoothing problems. We first plot the evolution of Dirichlet energy with node features propagated through the neural network in Figure 3. In addition, we evaluate our model performance with different depths of ACMP in Appendix D and compare it with GCN, GAT, GraphSAGE, and GRAND. The last baseline model is the representative GNN which is focused on limiting oversmoothing in message passing and uses a similar equation-driven propagation to ACMP.
>
> The code of our method and experiment examples has been provided in an anonymous repo with a website address in Appendix D. We added more comments on the code and demo recently and will publish the code in GitHub later.
>
> **Final comments:**
>
> Thanks for your feedback. We hope our explanation will be helpful to your questions. We emphasize the novelty of our model and clarify that ACMP is not just a simple utilization of the Allen-Chan model. We hope that, based on this, you will consider raising our score.

---

### Official Review · Reviewer_W9W7 · 2022-10-23

**Confidence:** 4
**Correctness:** 4
**Technical Novelty And Significance:** 3
**Empirical Novelty And Significance:** 3
**Recommendation:** 6

**Clarity, Quality, Novelty And Reproducibility:**

Aside from the comments in the previous section, clarity, quality and novelty are good enough. Regarding reproducibility, there is a link to an anonymous repo in the supplementary material pdf (although the code is mostly uncommented).

**Strength And Weaknesses:**

Strengths:
* The paper addresses an interesting problem in the area of deep learning for graphs, with an elegant approach
* The mathematical analysis seems sound and interesting

Weaknesses:
* Experimental settings are not fully convincing. In particular, from the manuscript it is not clear if hyperparameter tuning with model selection has been performed or not, and how this phase is conceived in order to ensure fairness in the performance assessment wrt literature methods. I could find some details in the supplementary material concerning the hyperparameters values explored for the proposed method (which sounds reasonable): I invite the authors to include this kind of information in the main text of the paper. Moreover, I suggest to clearly comment on the number of trainable parameters when comparing with the literature methods from the literature, in order to ensure that the performance advancements are not simply determined by a higher complexity of the resulting neural network (i.e., please report the number of trainable parameters in the proposed method vs those used in the experiments from the literature methods).
* Some aspects of the analysis are not fully clear. For instance, in the abstract it is claimed that "ACMP which has a simple implementation with a neural ODE solver can propel the network depth up to one hundred of layers", while I could not see any analysis in this sense in the paper. Moreover, while figure 2 is sufficiently explanatory and insightful, I could not find specific details on the network used to generate the plots. This, after all hampers the significance of the example, for which I suggest to include full details (at least in the caption).

**Summary Of The Paper:**

The paper falls in the area of deep learning for graphs. It is presented a new methodology for message passing, inspired from the study of interacting particles, in which the distinctive characteristic is that it has (in addition to conventional attracting forces) a repulsive force term in the message passing equation. This has the effect of improving the neural representations developed for heterophilic datasets and alleviate oversmoothing. Moreover, the proposed method includes an additional term called Allen-Cahn potential, to trying constraining the neural representations. The network architecture is run by a neural ODE solver, and it is demonstrated on several benchmarks.

Update after rebuttal:
I thank the authors for clarifying my doubts regarding the paper, and I am happy to confirm my score.

**Summary Of The Review:**

The paper presents an interesting methodology. In my view, downsides in the current version of the manuscript are mostly related to the experimental settings, where details about the model selection performed on the proposed model and on the competitors from the literature, do not completely eliminate doubts about the fairness of the reported experimental comparison.

---

> ### Author Response · Authors · 2022-11-16
> **Response to Reviewer W9W7**
>
> Thank you for your affirmation of our approach and mathematical analysis. Below we address each concern individually.
>
> *"Experimental settings are not fully convincing. In particular, from the manuscript, it is not clear if hyperparameter tuning with model selection has been performed or not, and how this phase is conceived in order to ensure fairness in the performance assessment wrt literature methods. I could find some details in the supplementary material concerning the values of the hyperparameters explored for the proposed method (which sounds reasonable): I invite the authors to include this kind of information in the main text of the paper. Moreover, I suggest clearly commenting on the number of trainable parameters when comparing with the literature methods from the literature, in order to ensure that the performance advancements are not simply determined by a higher complexity of the resulting neural network (i.e., please report the number of trainable parameters in the proposed method vs those used in the experiments from the literature methods)."*
>
> Thanks for your comments. We have added how hyperparameters are set in the main body, see Sec. 6. Due to the limitation of page length, we put the details of the hyperparameter search space in Appendix D.
>
> For the complexity of our model, we agree with your comment and we show both our model's trainable parameter numbers and the other benchmark model's trainable parameter numbers on the Cora dataset. As shown in the following table, ACMP is a lightweight neural network architecture that can achieve good classification performance with fewer parameters.
>
> |     Model | GCN  | GAT  | GraphSage | CGNN | GRAND-l | ACMP_GCN | ACMP_GAT |
> |------|------|------|-----------|------|---------|----------|----------|
> | Parameters | 144k | 230k | 200k      | 26k  | 17k     | 17k      | 19k      |
>
> *For "ACMP which has a simple implementation with a neural ODE solver can propel the network depth up to one hundred of layers":*
>
> ACMP is the solution of the particle system with pseudo-Ginzburg-Landau energy. We can directly implement ACMP in the GNN with any stable ODE solver (as described in Appendix D). We choose the Dormand-Prince5 method from the RK4 family as ACMP’s ODE solver, which has the best test performance among all solvers.
>
> *"while figure 2 is sufficiently explanatory and insightful, I could not find specific details on the network used to generate the plots."*
>
> Thanks for the comment. For Fig. 2 we use the initialization mentioned in Sec. 6.
> *The synthetic graph has 100 nodes with two classes and 2D feature which is sampled from the normal distribution with the same standard deviation $\sigma = 2$ and two means $\mu_1 = -0.5,$ $\mu_2 = 0.5.$ The nodes are connected randomly with probability $p = 0.9$ if they are in the same class, otherwise, nodes in different classes are connected with probability $p = 0.1.$*
> In Fig. 2, we visualize how the node features evolve from their initial state to their final steady state when 50 layers of GNN are applied. We have added that explanation in the revision.
>
>
> **Clarity, Quality, Novelty And Reproducibility:**
>
> *Aside from the comments in the previous section, clarity, quality and novelty are good enough. Regarding reproducibility, there is a link to an anonymous repo in the supplementary material pdf (although the code is mostly uncommented).*
>
> Thanks for your comments on our work’s clarity, quality, and novelty and your kind advice on our code. We have added more comments recently and will soon put it out with the best parameters to guarantee reproducibility.

---

### Official Review · Reviewer_A1JY · 2022-10-25

**Confidence:** 4
**Correctness:** 3
**Technical Novelty And Significance:** 4
**Empirical Novelty And Significance:** 2
**Recommendation:** 6

**Clarity, Quality, Novelty And Reproducibility:**

The current logic in Sec 1 is not clear. I would suggest the authors changing to sth. like the following:
1. We summarize the existing GNNs into a unified viewpoint, i.e., they are learning with the Dirichlet energy.
2. But this energy has xxx drawbacks, e.g., oversmoothing.
3. We propose using AC energy, with xxx benefits. Then with AC energy, we propse the ACMP.

I would like to raise the score after the authors fix this issue.


**Strength And Weaknesses:**

## Strengths
- This paper introduces using an interacting particle system with attractive and repulsive forces and the Allen-Cahn force for message passing.
- The Sec 3 & 4 are technically sound and clearly explained.
## Weakness
- The motivation is missing, and why we want to study Allen-Cahn message passing (ACMP) is not explained.
  - In Sec 1, the authors mention the that particle system is common in nature and human society, but the difference between ACMP and traditional GNN is not clearly explained.
  - Namely, a smoother logic is that, what is missing in the current GNN message passing, and how this can motivate ACMP.
  - The oversmoothing should be briefly introduced in Sec 1. Now it just shows up directly in paragraph 1 and 2, making it hard to follow, e.g., how the repulsive orces can avoid oversmoothing is not clear.

I need to confirm some points in Sec 1 with the authors:
- For sentence “Most existing message passing neural networks are driven by attractive forces associated with the Dirichlet energy …” Can authors give more detailed explanations?
- What is the difference between Allen-Cahn energy and Dirichlet energy? So that AC energy can avoid the exploding issue when the GNN gets deeper?

Some questions on Sec 2.
  - There exist several deep GNN works [1, 2], but not discussed in the paper. They are also related to the oversmoothing. Besides, I’m wondering which category would they fall into the framework introduced by the authors.
  - Before Eq(2), the authors say ‘’ (GRANND) … for some message passings:’’, can authors add citations for these message passing works?




- I have two concerns on the experiments.
  - The performance on existing datasets are far behind the SOTA, e.g., https://paperswithcode.com/sota/node-classification-on-cora. Can authors also switch to some of the SOTA GNNs in the board?
  - A minor gap between method and experiments. Since ACMP can mimic the particle dynamics, the authors should conduct experiments on more related datasets, like quantum chemistry dataset. The current experiments are on the datasets like Cora, where the notions like forces and potential are hard to verify.


[1] Gallicchio, Claudio, and Alessio Micheli. "Fast and deep graph neural networks." Proceedings of the AAAI conference on artificial intelligence. Vol. 34. No. 04. 2020.

[2] Yan, Yujun, et al. "Two sides of the same coin: Heterophily and oversmoothing in graph convolutional neural networks." arXiv preprint arXiv:2102.06462 (2021).


**Summary Of The Paper:**

This paper proposes a novel message passing algorithm for graph neural networks. It explores both the attractive and repulsive forces simultaneously. Intuitively, it is introducing a measure to tell when to gather/diverge the neighborhood nodes.


**Summary Of The Review:**

I think this paper is technically interesting, and I can understand the core ideas of the authors. I have two major concerns:
1. The presentation is not clear, especially Sec 1.
2. The experiments are a little weak. Such particle system can be better verified on some quantum datasets. If the results still hold, then I think this would be a very solid paper.

---

> ### Author Response · Authors · 2022-11-16
> **Thank you for your review: rebuttal part 1**
>
> Thank you for the detailed feedback and for valuable comments. We address them below one by one and please kindly let us know if there are further doubts or clarifications required.
>
> *"The motivation is missing, and why we want to study Allen-Cahn message passing (ACMP) is not explained.
> In Sec 1, the authors mention that particle system is common in nature and human society, but the difference between ACMP and traditional GNN is not clearly explained."*
>
> Generally, we want to introduce a view of particle systems to analyze and design neural message passing. ACMP is our first attempt to apply the formulation. This is our basic motivation.
>
> - The design philosophy of ACMP is different from traditional GNN. To our knowledge, the theoretical motivations of traditional GNNs focus on graph convolutions, spectral graph theory, graph attention mechanism, and differential equations but are limited to macroscopic physics (like GRAND inspired by the diffusion process and GraphCON by non-linear oscillators). In contrast, our ACMP model stems from a microscopic physics view. Using microscopic particle dynamics we can flexibly and elaborately design the dynamic behaviors by introducing the terms such as attraction and repulsion between particles, which the systems of macroscopic physics cannot achieve.
> - As we mentioned in Sec 1, nodes and edges in a graph can be viewed as particles and the interaction between them. This helps design a microscopic-physically-inspired approach. Our approach is focused on feature dynamics in the scale of particle systems rather than diffusion equations in hydromechanics.
> - ACMP reflects the dynamics of all nodes. Its evolution and interplay of particles of the graph are very different from existing message passings. One can see it by comparing equation (5) in the paper with the updated formulations of
>   - GCN: $\mathbf{x}^{'}_i = \mathbf{\Theta}^{\top} \sum\_{j \in\mathcal{N}\_i \cup \{i\}} \frac{a\_{j,i}}{\sqrt{\hat{d}\_j  \hat{d}\_i}} \mathbf{x}\_j,$
>   - and GAT: $\mathbf{x}^{'}\_i = \alpha\_{i,i}\mathbf{\Theta}\mathbf{x}\_{i} + \sum\_{j \in \mathcal{N}\_i} \alpha\_{i,j}\mathbf{\Theta}\mathbf{x}\_{j}.$
>   - ACMP can explicitly express not only attraction between neighbors but also repulsion, which is the key point to enhance its effectiveness.
>
> *"Namely, a smoother logic is that what is missing in the current GNN message passing, and how this can motivate ACMP."*
>
> Yes. It is a smooth logic to show oversmoothing and heterophily as motivations for ACMP. We will clarify this point in Sec 1 although the discussion in Sec 3.1 mentioned a few on it.
> Nevertheless, please note that the repulsion and potential mechanism in ACMP is not just a special circumvention for oversmoothing and heterophily but a complement for the precious attraction-only message-passing mechanism. That is what we want to emphasize.
>
> *"The oversmoothing should be briefly introduced in Sec 1. Now it just shows up directly in paragraphs 1 and 2, making it hard to follow, e.g., how the repulsive forces can avoid oversmoothing is not clear."*
>
> Thanks for your advice. We are glad to explain more on the logic in Sec 1 (revision edition).
> - First, we introduced a perspective of particle systems to analyze message passings in GNNs. In this view, we proposed ACMP as a novel message-passing model employing interacting particle dynamics. Then we explained the significance of each part of ACMP.
> - In paragraph 2, we showed that the structure of ACMP has nice properties:
>   - Circumvents oversmoothing
>   - Works on heterophilic graphs
> - In paragraph 3 and 4, we summarize the major features of ACMP and our theoretical results.
>
> About *"how the repulsive forces can avoid oversmoothing,"* we explained it in the first paragraph in Sec 5. Here we would like to explain more: Oversmoothing can be viewed as the phenomenon that features become indistinguishable as GNN deepens. In the context of particle systems, it means that all the particles aggregate into a **consensus** and an attractive-only system leads to such a convergence. Hence, modifying the system by adding repulsive force is useful for limiting oversmoothing.

---

> ### Author Response · Authors · 2022-11-16
> **Thank you for your review: rebuttal part 2**
>
> About *"how the repulsive forces can avoid oversmoothing,"* we explained it in the first paragraph in Sec 5. Here we would like to explain more: Oversmoothing can be viewed as the phenomenon that features become indistinguishable as GNN deepens. In the context of particle systems, it means that all the particles aggregate into a **consensus** and an attractive-only system leads to such a convergence. Hence, modifying the system by adding repulsive force is useful for limiting oversmoothing.
>
> *"For sentence “Most existing message passing neural networks are driven by attractive forces associated with the Dirichlet energy …” Can authors give more detailed explanations?"*
>
> Here examples include GCNConv, GATConv, GINConv, ChebConv, GRAND etc. Take GCN and GAT as examples. Intuitively, observe the update formulas:
> - GCN: $\mathbf{x}^{\prime}\_i = \mathbf{\Theta}^{\top} \sum\_{j \in
>  \mathcal{N}\_i \cup \{ i \}} \frac{a\_{j,i}}{\sqrt{\hat{d}\_j \hat{d}\_i}} \mathbf{x}\_j,$
> - GAT: $\mathbf{x}^{\prime}\_i = \alpha\_{i,i}\mathbf{\Theta}\mathbf{x}\_{i} +
>     	\sum\_{j \in \mathcal{N}\_i} \alpha\_{i,j}\mathbf{\Theta}\mathbf{x}\_{j},$
>
> We can see that all the coefficients (like $\alpha_{i,j},$ $\mathbf{\Theta}$)
> are nonnegative, which means that the effect from $x_j$ (or $x_i$) to $x_i$ is to make $x_i$ similar to $x_j$ (or $x_i$). And that is the attractive force in your question.
>
> Also, under certain conditions, the update formulas can be viewed as numerical schemes of odes.
> We ignore the learnable parameters $\Theta,$ i.e.
> - GCN: $\mathbf{x}^{\prime}\_i = \sum\_{j \in
>  \mathcal{N}\_i \cup \{ i \}} \frac{a\_{j,i}}{\sqrt{\hat{d}\_j \hat{d}\_i}} \mathbf{x}\_j,$
> - GAT: $\mathbf{x}^{\prime}\_i = \alpha_{i,i}\mathbf{x}\_{i} +
>     	\sum\_{j \in \mathcal{N}\_i} \alpha\_{i,j}\mathbf{x}\_{j}.$
>
> GAT can be written as
>
> $$\mathbf{x}^{\prime}_i - \mathbf{x}_{i} =
>     	\sum\_{j \in \mathcal{N}\_i \cup\{i\}} \alpha\_{i,j}(\mathbf{x}\_{j}-\mathbf{x}\_{i}).$$
>
> This is the forward Euler discretization of the ODE
>
> $$\frac{d}{dt}\mathbf{x}\_{i} = \sum\_{j \in \mathcal{N}\_i \cup\{i\}} \alpha\_{i,j}(\mathbf{x}\_{j}-\mathbf{x}\_{i}),$$
>
> which is an attractive-only system as Sec 3.1 discusses.
>
> Similarly, for a graph with $d\_i=d\_j$ for all $i,j$ (e.g. a fully connected one), GCN becomes
>
> $$\mathbf{x}^{\prime}\_i - \mathbf{x}\_{i} = \sum\_{j \in \mathcal{N}\_i \cup\{i\}} \frac{a\_{j,i}}{\sqrt{\hat{d}\_j \hat{d}\_i}} (\mathbf{x}\_{j}-\mathbf{x}\_{i}),$$
>
> which is the forward Euler discretization of a slightly different ODE
>
> $$\frac{d}{dt}\mathbf{x}\_{i} = \sum\_{j \in \mathcal{N}\_i \cup\{i\}} \frac{a\_{j,i}}{\sqrt{\hat{d}\_j \hat{d}\_i}} (\mathbf{x}\_{j}-\mathbf{x}\_{i}).$$
>
> The attraction will bring about the Dirichlet energy to decay to zero very quickly as shown in Fig 3.
>
> *"What is the difference between Allen-Cahn energy and Dirichlet energy? So that AC energy can avoid the exploding issue when the GNN gets deeper?"*
>
> Mathematically, Allen-Cahn energy (or more appropriately, pseudo-Ginzburg-Landau energy, as we now term in the paper to replace Allen-Cahn energy) and Dirichlet energy are explicitly given in Eq.(8) Sec~3.2 (in the revised submission) and Eq.(6). There are two main differences between the two energies:
> - The existence of $\beta_{i,j}$ in the first term of $\Phi$ makes it just a pseudo energy, which means that $\Phi$ might be nonpositive for some $\mathbf{x}$. But one benefit lies in that it can derive particle dynamics with both attractive and repulsive forces. Dirichlet energy is true nonnegative energy and can only derive an attractive system under gradient flow. So the Dirichlet energy can be used as the gradient flow and also the measure for feature difference. But the pseudo-Ginzburg-Landau energy in our paper is given as a gradient-flow explanation of ACMP to help understand and inspire energy-based dynamics design. It cannot be used as a measure.
>
> "Pseudo-Ginzburg-Landau energy can avoid exploding" is roughly right. Exactly, it is the potential $W(x)$ that avoids the infinite separation of features due to repulsion. It is allowable to use other kinds of potentials as we mentioned in the appendix. But we think the double-well potential is a classical choice with physical significance (coordinating with the Allen-Cahn model or Cucker-Smale model).

---

> ### Author Response · Authors · 2022-11-16
> **Thank you for your review: rebuttal part 3**
>
> *"There exist several deep GNN works [1, 2], but not discussed in the paper. They are also related to the oversmoothing. Besides, I’m wondering which category would they fall into the framework introduced by the authors."*
>
> Thank you for the references. [2] introduces signed messages to analyze heterophily and oversmoothing problems. The separation of positive and negative matrices in GGCN [2] shares a similar form with our attraction-repulsion design. Both of them emphasize the effect of “negative message”. However, they have different theoretical views and model construction. GGCN focuses on improving the original GCN structure, and ACMP focuses on a new model structure based on the dynamics of interacting systems. [1] proposed Fast and Deep Graph Neural Network (FDGNN) by exploiting the fixed point of the recursive system to embed the inputs. It is a different model focusing on embedding while our model focuses on message passing. We have included them in the revised edition.
>
>
> *"Before Eq(2), the authors say ‘’ (GRANND) … for some message passings:’’, can authors add citations for these message passing works?"*
>
> Thank you. The test says “Neural diffusion equations on graphs (GRAND) are proposed by Chamberlain et al. (2021), which provides a unified mathematical framework for some message passings” Here, “some message passings” means message passings like GRAND-l, GRAND-nl, GRAND-nl-rw and other message passings applying different numerical schemes for Eq.(2) in Chamberlain et al. (2021). They are sorts of GRAND actually (GRAND architectures are a class of GNNs based on Eq. (2)).
>
> Benjamin Paul Chamberlain, James Rowbottom, Maria I. Gorinova, Stefan D Webb, Emanuele Rossi, and Michael M. Bronstein. GRAND: Graph neural diffusion. In ICML, 2021. URL https://openreview.net/forum?id=_1fu_cjsaRE
>
> *"I have two concerns about the experiments.
> The performance on existing datasets is far behind the SOTA, e.g., https://paperswithcode.com/sota/node-classification-on-cora. Can authors also switch to some of the SOTA GNNs in the board?"*
>
> We used a different splitting dataset method from those SOTA methods listed in Paperwithcode. As (Shchur et al., 2018) mentioned, a single split of training/validation/test split will give fragile and misleading results. To address the limitations of this evaluation methodology, we apply 100 random splits with 10 random initializations into all *Homophilic* datasets which we report in Sec.6. This can overcome the limitation, but testing one model requires a lot of time. In this case, we pick some SOTA models and report them in our paper.
>
> In addition, ACMP can be treated as a trick to improve those SOTA methods. One can use all these SOTA methods to compute the ‘message’ $a_{i,j}$ in Eq 9. By adding the repulsive force $\beta$ and double well potential and fine-tuning, we find it will improve the performance on a different dataset. It has slight improvements for homophilic datasets such as the Cora dataset due to the high homophily level of the Cora dataset and the smoothing with neighbours makes a difference for the model. Conversely, smoothing with neighbours will cause poor performance for heterophilic datasets. The addition of repulsive forces $\beta$ and double-well potentials improves model performance significantly.
>
> Shchur, O., Mumme, M., Bojchevski, A., and Günnemann, S. Pitfalls of graph neural network evaluation. arXiv:1811.05868, 2018.
>
> *"A minor gap between method and experiments. Since ACMP can mimic particle dynamics, the authors should conduct experiments on more related datasets, like the quantum chemistry dataset. The current experiments are on the datasets like Cora, where the notions like forces and potential are hard to verify."*
>
> The particle system mechanism is originally suitable for node classification tasks. For graph-level tasks, including PPI and ZINC, ACMP can be interpreted as a feature extraction that provides different modes of each graph or molecule.
>
> Protein-Protein Interactions (PPI):
> | model             | GCN  | GAT  | GraphSAGE | PDE-GCN | GCNII | GeniePath | JKNet | ACMP_GCN |
> |:-------------------:|:------:|:------:|:-----------:|:---------:|:-------:|:-----------:|:-------:|:----------:|
> | Micro-averaged F1 | 98.5 | 97.3 | 61.2      | 99.2    | 99.5  | 98.5      | 97.6  | 99.3     |
>
> ZINC (small 12k version)
> | model | GCN            | GAT            | GIN            | GatedGCN       | GraphSAGE      | PNA            | ACMP_GCN       |
> |:-------:|:----------------:|:----------------:|:----------------:|:----------------:|:----------------:|:----------------:|:----------------:|
> | MAE   | 0.47$\pm$0.002 | 0.46$\pm$0.002 | 0.41$\pm$0.008 | 0.42$\pm$0.006 | 0.41$\pm$0.005 | 0.32$\pm$0.032 | 0.23$\pm$0.008 |

---

> ### Author Response · Authors · 2022-11-16
> **Thank you for your review: part 4**
>
> *For "Clarity, Quality, Novelty And Reproducibility:"*
>
> Thanks for the suggestion which logic makes sense.
>
> The gist of the paper is beyond the oversmoothing and energy. We introduced a new GNN model based on interacting  particle systems and showed the effect of the attraction-repulsion system by designing ACMP. The alleviation of oversmoothing and the effect on heterophilic datasets support our emphasis on such dynamical mechanism.
>
> We want to explain more about ACMP and the Dirichlet energy. ACMP is a model mainly motivated by attraction-repulsion dynamic systems. It focuses on feature dynamics during the message passing process. Furthermore, by discussing the interaction dynamics, we propose pseudo-Ginzburg-Landau energy to generalise the model and modify it by adding potential in terms of gradient flow theory. The Dirichlet energy, under the formulation of gradient flow, can derive an attraction-only system (7). But certainly (7) has drawbacks on dealing with heterophilic datasets and circumventing oversmoothing, as we mentioned in Sec 3.
>
> However, the Dirichlet energy is still a useful measure in oversmoothing. We use it to illustrate ACMP can circumvent oversmoothing in Sec 5. (See also our reply for the previous question related to the organization of Sec. 1.)
>
> **Final comment:** Thanks for the feedback and the many comments and questions which we believe have helped to improve the revised version.  We hope our explanation will be helpful for your questions.

---

> > ### Comment · Reviewer_A1JY · 2022-11-18
> > **Thank you for the reply**
> >
> > The authors have addressed my main concerns during the rebuttal.
> >
> > - Generally, I think this paper is interesting in introducing the particle system for handling structure data modeling.
> > - After the latest revision, the motivation is reasonable for readers.
> > Thus, I would like to raise the score.
> >
> > Meanwhile, the empirical results can be more convincing if more solid tasks are considered. I'm aware of the existing literature, where many papers are only discussing tasks like Cora etc, and results can be quite tricky using different seeds or other hyperparameters. From this aspect, the particle system perspective proposed in this paper can be more interesting if it shows some qualitative results using physics tools. (Now only the quantitative improvements are available.)

---

> > > ### Author Response · Authors · 2022-11-18
> > > **Thank you for your advice**
> > >
> > > We are grateful for your advice. We will conduct more experiments on different tasks in the improvement of the paper. And we will try to apply physics tools to illustrate the quantitative results in the future.

---

### Author Response · Authors · 2022-11-16
**Thanks for the reviews: general comment + list of revisions**

**General：**

We are grateful for the reviewers for their feedback and comments. We have made our strength to address these, see below. We are also grateful for agreeing on the major merits of our submission. Here, we comment on the points that are commonly focused on by all reviewers. We will then address any question and/or comment in detail.

**What's our main contribution?**

In this paper, our main contribution is to establish a new GNN model under the formulation of particle systems which consequently has merits that the most existing GNNs do not take. ACMP combines the attraction-repulsion dynamical mechanism with a damping term induced by the double-well potential (the combined energy which gradient flow induces the equation of the message passing becomes pseudo-Ginzburg-Landau energy) is then and works as a useful message passing module.

The advantage of Allen-Cahn message passing with repulsion is threefold:
- It does not have oversmoothing issue, namely the Dirichlet energy is bounded from below.
- The network is stable in the sense that features and Dirichlet energy is bounded from above.
- Feature smoothness (or energy decrease) and the balance between nodes features and graph features can be adjusted via three parameters alpha, beta and delta.

In theory, we prove that the Dirichlet energy of GNNs with ACMP has a lower bound (no oversmoothing), as well as an upper bound (no blowing up) under specific conditions. This agrees with the experimental results shown in Fig.3. We prove that ACMP is a process for the features to generate clusters, which can be an interpretation for classification tasks.

We can achieve SOTA performance in multiple homophilic and heterophilic datasets by using one-hop neighbour information only. (Note that the evolution of the particles for ACMP will involve the use of multi-hop information.)

**What's new in the modified version of the manuscript? (The modified part is highlighted in blue.)**

In light of feedback from the reviewers the submission has been revised as follows:
- We modified the structure in Sec.1 and emphasised the motivation and advantages of our model AMCP.
- To avoid confusion, we replaced the “Allen-Cahn potential” and “” by “double-well potential”.
- We added more instruction of Figure 2 in the main body.
- We added some references on oversmoothing and deep GNNs.
- We added a parameter comparison for GCN,GAT, GraphSage, CGNN, GRAND-l and our models in Appendix D.2.

*We address each of the reviews individually below.*

---

### Decision · Program_Chairs · 2023-01-20

**Decision:**

Accept: notable-top-25%

**Justification For Why Not Higher Score:**

The reviewers' consensus is for 'weakly accepting' the paper. I felt that suggesting it be elevated to 'Spotlight' level already goes somewhat against this consensus, and also I do not find this paper seminal enough, or interesting enough to wide range of ICLR audiences, to be considered for a long oral.

**Justification For Why Not Lower Score:**

I am choosing to nominate this paper for a Spotlight presentation. It proposes a very timely piece of theory, which seems to line up quite nicely for a very important problem in graph representation learning (heterophilous datasets). The write-up is solid and the results are exactly what I would expect for a paper like this. I believe it should be given more 'screen-time' at the conference, to increase the work's reach -- because there is a good chance it might be useful broadly in the graph representation learning community.

**Metareview: Summary, Strengths And Weaknesses:**

In this work, the authors propose the Allen-Cahn message passing operator, which is capable of expressing both attractive and repulsive forces between the nodes. This is a highly suitable development for heterophilous datasets, because this is _exactly_ what a GNN needs to do in such a setting: choose which of the neighbouring nodes are positive for the classification prediction, in a setting where most of them are negative. The chosen experimental tasks are relevant for validating this hypothesis, and the resulting ACMP layer appears to deliver on this promise. The reviewers raised relevant points in their reviews, to which the authors satisfactorily responded, resulting in a consensus for acceptance. I concur with the reviewers, and support publication of this paper.

**Note From Pc:**

if the above contains the word "oral" or "spotlight" please see: "oral" presentation means -> notable-top-5% and "spotlight" means -> notable-top-25%. As stated in our emails, we are disassociating presentation type from AC recommendations